# A Warburg-like metabolic program coordinates Wnt, AMPK, and mTOR signaling pathways in epileptogenesis

**Roaya S. Alqurashi**[1,2,3☉], **Audrey S. Yee**[4☉¤], **Taylor Malone**[1‡], **Sumaiah Alrubiaan**[1,2,3‡], **Mary W. Tam**[1,5‡], **Kai Wang**[1,5], **Rozena R. Nandedwalla**[1,5], **Wesley Field**[1], **Dalal Alkhelb**[1,2,6], **Katherine S. Given**[7], **Raghib Siddiqui**[5], **James D. Baleja**[1,2], **K. Eric Paulson**[1,2], **Amy S. Yee**[1,2]*

1 Department of Developmental, Molecular and Chemical Biology, Tufts University School of Medicine, Boston, Massachusetts, United States of America, 2 Graduate Program in Pharmacology and Experimental Therapeutics, Graduate School of Biomedical Sciences, Tufts University School of Medicine, Boston, Massachusetts, United States of America, 3 Pharmacology and Toxicology Department, College of Pharmacy, Umm Al-Qura University, Makkah, Saudi Arabia, 4 Neurology Section, Rocky Mountain Regional VA Medical Center, Aurora, Colorado, United States of America, 5 Master's Program in Biomedical Sciences, Tufts University School of Medicine, Boston, Massachusetts, United States of America, 6 Pharmacology and Toxicology Department, College of Pharmacy, King Saud University, Riyadh, Saudi Arabia, 7 Department of Cell and Developmental Biology, University of Colorado Anschutz Medical Campus, Aurora, Colorado, United States of America

☉ These authors contributed equally to this work.
¤ Current address: National Institute of Allergy and Infectious Disease, Neuroimmunological Disease Section, Bethesda, Maryland, United States of America
‡ TM, SA and MWT also contributed equally to this work. JDB, KEP and ASY are joint senior authors on this work.
* amy.yee@tufts.edu

**Data Availability Statement:** All relevant data are within the paper and its Supporting information files.

## Abstract

Epilepsy is a complex neurological condition characterized by repeated spontaneous seizures and can be induced by initiating seizures known as status epilepticus (SE). Elaborating the critical molecular mechanisms following SE are central to understanding the establishment of chronic seizures. Here, we identify a transient program of molecular and metabolic signaling in the early epileptogenic period, centered on day five following SE in the pre-clinical kainate or pilocarpine models of temporal lobe epilepsy. Our work now elaborates a new molecular mechanism centered around Wnt signaling and a growing network comprised of metabolic reprogramming and mTOR activation. Biochemical, metabolomic, confocal microscopy and mouse genetics experiments all demonstrate coordinated activation of Wnt signaling, predominantly in neurons, and the ensuing induction of an overall aerobic glycolysis (Warburg-like phenomenon) and an altered TCA cycle in early epileptogenesis. A centerpiece of the mechanism is the regulation of pyruvate dehydrogenase (PDH) through its kinase and Wnt target genes PDK4. Intriguingly, PDH is a central gene in certain genetic epilepsies, underscoring the relevance of our elaborated mechanisms. While sharing some features with cancers, the Warburg-like metabolism in early epileptogenesis is uniquely split between neurons and astrocytes to achieve an overall novel metabolic reprogramming. This split Warburg metabolic reprogramming triggers an inhibition of AMPK and subsequent activation of mTOR, which is a signature event of epileptogenesis. Interrogation of the

**Funding:** This work was supported by grants from the Department of Defense ((W81XWH-10-1-0381; ASY and ASY), the CURE Foundation (www. cureepilepsy.org;ASY and ASY) and by Umm Al-Qura University, Makkah, Saudi Arabia (RA). The work utilized NMR instrumentation that was purchased with funding from a National Institutes of Health SIG grant (S10OD020073). Please note that the funders had no role in study design, data collection and analysis, decision to publish, or preparation of the manuscript. Their contribution was nonetheless instrumental to this project.

**Competing interests:** The authors have declared that no competing interests exist.

mechanism with the metabolic inhibitor 2-deoxyglucose surprisingly demonstrated that Wnt signaling and the resulting metabolic reprogramming lies upstream of mTOR activation in epileptogenesis. To augment the pre-clinical pilocarpine and kainate models, aspects of the proposed mechanisms were also investigated and correlated in a genetic model of constitutive Wnt signaling (deletion of the transcriptional repressor and Wnt pathway inhibitor HBP1). The results from the HBP1$^{-/-}$ mice provide a genetic evidence that Wnt signaling may set the threshold of acquired seizure susceptibility with a similar molecular framework. Using biochemistry and genetics, this paper outlines a new molecular framework of early epileptogenesis and advances a potential molecular platform for refining therapeutic strategies in attenuating recurrent seizures.

## Introduction

Epilepsy is a chronic neurological condition affecting around 1% of the global population and is characterized by recurrent spontaneous seizures with potential comorbidities [1, 2]. Epileptogenesis is generally initiated by an inciting event, such as a prolonged seizure called status epilepticus (SE), brain tumors, head injury, infection, or other causes. The initiating event in epileptogenesis is followed by a "latent period", which can last from weeks to years, and then is followed by the subsequent emergence of spontaneous seizures. During this latent period, there are numerous and to-be-defined cellular and molecular changes that lead to the emergence of clinically significant, spontaneous seizures that defines full-blown epilepsy (e.g. [3]).

In this paper, we investigate the complex molecular changes that occur in early epileptogenesis with the objective of discovering therapeutic targets. A clue was the observation that the mammalian target of rapamycin (mTOR) was transiently activated in the hippocampus by kainate around 5 days post-status epilepticus (SE) [4], so we reasoned that the molecular signaling network regulating mTOR will have an important role in driving epileptogenesis to epilepsy. The mTOR pathway has been implicated in epilepsy in many human and mouse studies. For example, loss of Tuberous Sclerosis Complex (TSC1) or (TSC2), which are upstream regulators of the mTOR pathway, resulted in constitutive neuronal mTOR activity and acquired seizures [5–7]. Induced models of temporal lobe epilepsy (TLE) also demonstrate increased mTOR signaling [4, 8–10]. Treatment of mouse models of TSC inactivation with the mTOR inhibitor rapamycin both reduced the number of seizures and increased mouse survival [11–13]. Similarly, rapamycin reduced seizures in a mouse model of traumatic brain injury-induced epilepsy and infantile spasms [14–16]. Lastly, the mTORC1 inhibitor everolimus, initially used in clinical trials for cancers, resulted in a >50% reduction in seizure frequency in a small cohort of TSC patients [17]. Similar results were demonstrated with the related mTORC1 inhibitor sirolimus [18].

The mTOR pathway itself integrates multiple signaling pathways, including phosphatidylinositol 3-kinase (PI3K) and Wnt signaling [19]. Wnt signaling is activated by Wnt ligands binding to 7-membrane frizzled receptor (FZD). The binding of FZD results in the activation of disheveled (Dvl) and inhibits glycogen synthase kinase 3 beta (GSK3β) via phosphorylation. The phosphorylated and inactivated GS3K3β results in un-phosphorylated and stabilized β-catenin in the cytoplasm. The stabilized β-catenin then translocates into the nucleus and binds to the T-cell factor/lymphoid enhancer factor-1 (TCF/LEF1) transcription factors. This event then leads to the activation of a unique Wnt-signaling driven gene expression program that establishes cellular consequences that are unique to tissue or disease state. Other experimental

evidence suggests activation of mTOR via conditional knockout of phosphatase and tensin homolog (PTEN) was sufficient to induce seizures [20]. Inactivation of PTEN resulted in PI3K activation and subsequent mTOR induction. Molecular studies in cancer showed that Wnt signaling controlled the activity of the mTOR pathway via GSK3β phosphorylation of TSC2, promoting the inhibitory function of TSC2 on mTOR activity [21]. Functionally, GSK3β phosphorylation of TSC2 is primed by 5' adenosine monophosphate-activated protein kinase (AMPK) phosphorylation of TSC2. The combination of GSK3β and AMPK phosphorylation thus fully stimulates the activity of TSC2 and inhibition of mTOR. TSC2 thus integrates Wnt signaling and cellular energy status into the mTOR pathway. The functional connection between the Wnt and mTOR pathways was further demonstrated in experiments where the growth of Wnt1-driven mammary tumors was attenuated by rapamycin [21]. Interestingly lithium, an inhibitor of GSK3β, can potentiate seizure activity induced by the pilocarpine model of TLE [22], although the mechanism for this work was not determined. In cancer and developmental biology studies, lithium chloride is often used to activate Wnt signaling through its actions on GSK3β [23]. A schematic figure to summarize how TSC2 functions to integrate signals into mTOR is shown in Fig 1A.

The functional intersection of Wnt and mTOR signaling in other diseases prompted us to investigate whether Wnt signaling contributes to seizure pathology. Interestingly, both Wnt and mTOR signaling were possibly associated with dysregulated microRNA (miRNA) expression in different hippocampal subregions of experimental TLE at day 1, day 7, and chronic seizures after SE [31]. Wnt signaling has a well-defined role in brain and in early hippocampal development [32–35] and remains crucial for postnatal hippocampal function. A perinatal conditional knockout of β-catenin showed significant hippocampal defects, particularly in the dentate gyrus (DG), suggesting an important role for β-catenin and Wnt signaling in development [36]. The proliferation of adult hippocampal progenitor cells is dependent upon Wnt signaling. Both Wnt ligands and frizzled receptors are expressed in the subgranular zone (SGZ) of the DG (reviewed in [35]), and lineage-tracing experiments demonstrate that long-lived Wnt-β-catenin-responsive stem cells reside in the DG [33]. Injection of lentiviral-expressed dominant negative Wnt1 or functional Wnt3 provided direct evidence that Wnt signaling regulates neurogenesis in the DG [37]. Lastly, ectopic expression of a β-catenin/LEF fusion protein in the cortex drives the development and differentiation of hippocampal-like cells. Because the LEF/TCF transcription factors are the downstream recipients of Wnt signaling and establish the phenotype through activated gene expression, the β-catenin/LEF fusion protein in the above context would mimic excessive Wnt-signaling-driven gene activation by these critical transcription factors [36].

Given the importance of Wnt signaling in both hippocampal development and function of the established molecular integration of Wnt and AMPK signaling into mTOR activity (in cancer cells [24–26]), we investigated whether Wnt and AMPK signaling were altered in early epileptogenesis and could regulate the mTOR activation previously established in epileptogenesis. We discovered that Wnt signaling also has a metabolic consequence in the hippocampus to coordinate a Warburg metabolism, which had functional consequences downstream into AMPK-to-mTOR signaling. At its simplest, Warburg metabolism results in increased glycolysis with pyruvate conversion to lactate under aerobic conditions [25, 38]. The molecular model we developed was tested by both pharmacological and genetic experiments. Together, the biochemical, cellular, pharmacological and genetic approaches suggest that Wnt signaling and its associated network may set the threshold for seizure susceptibility and onset. These diverse studies provide a molecular framework to include Wnt and metabolic signaling via AMPK into the signaling network governing mTOR activation in early epileptogenesis. These studies implicate new possibilities for therapeutic targeting in epileptogenesis.

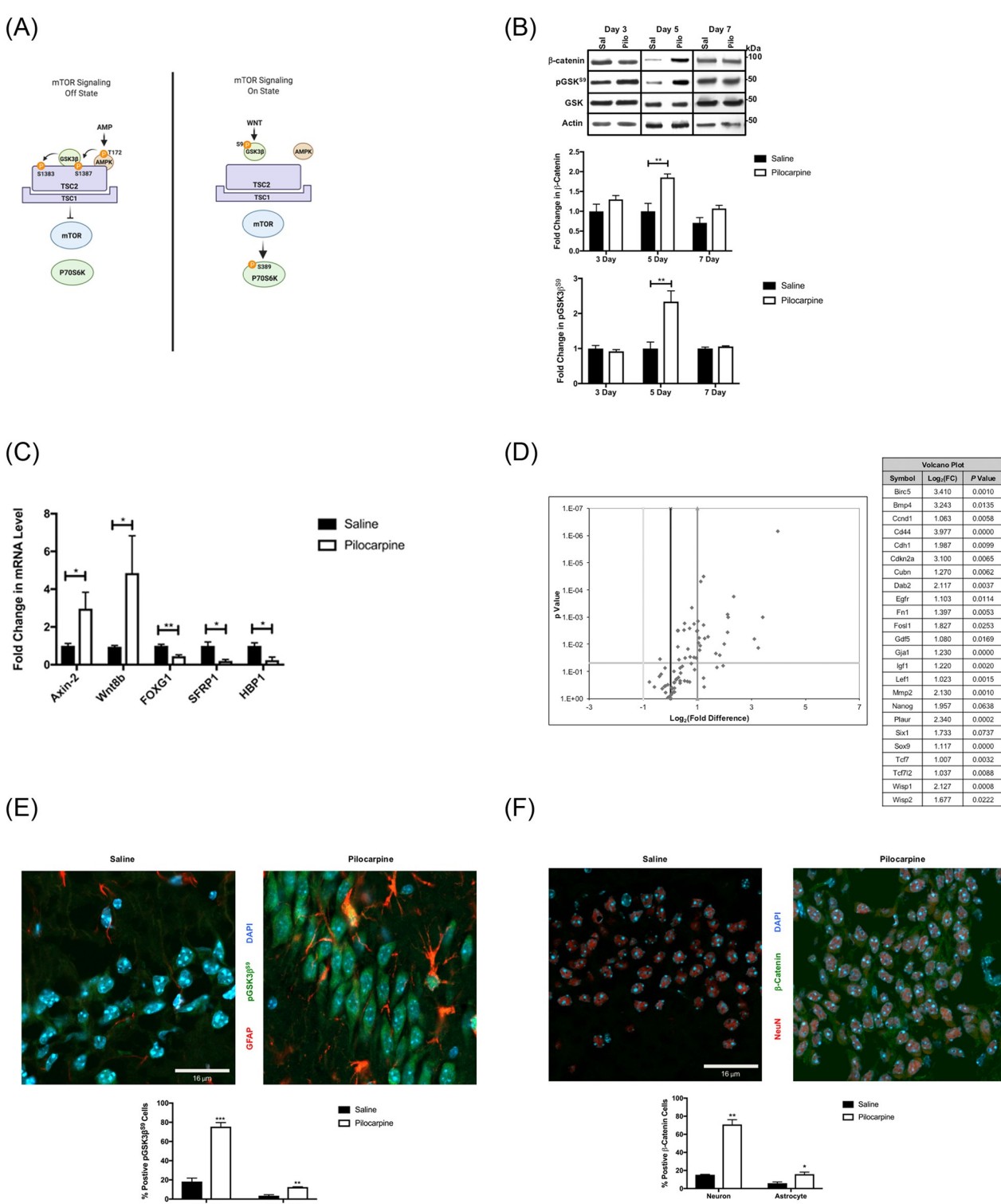

**Fig 1. Status epilepticus (SE)-induced Wnt signaling and associated signaling networks.** (A) Model of Wnt signaling and mTOR activation (via TSC2 and AMPK). TSC2 requires the phosphorylation of both AMPK and GSK3β$^{S9}$ to fully inhibit the downstream activity of mTOR pathway, including reduced phosphorylation of P70S6K. The figure is based on the literature [21, 24–26] and was created with BioRender.com. (B) Wnt signaling is activated in the early epileptogenic period (pilocarpine induction). Status epilepticus was induced in male 7-week FVBN mice with pilocarpine (220 mg/kg). Representative western blots and quantified data of hippocampal extracts from 3-, 5- or 7-day post-pilocarpine (Pilo; n = 4) or control saline (Sal; n = 4) are shown for pGSK3β$^{S9}$, GSK3β, β-catenin, and actin (loading control). (C). Expression of selected Wnt pathway targets

and components reflects increases in Wnt signaling. Hippocampal mRNA levels for representative Wnt pathway target gene (Axin 2), a Wnt ligand (Wnt8B), and Wnt pathway inhibitors sFRP1 [27, 28] and HBP1 [29, 30] were measured by quantitative RT-PCR at day 5 post-SE (pilocarpine) compared to control mice. (n = 4–6 mice for each group). The data were represented as mean ± SEM using unpaired student's *t*-test. (Prism 9.0, Graphpad. $^*P < 0.05$, $^{**}P < 0.01$). (D) Wnt target genes are activated in the early epileptogenic period. A qRT- PCR based array of representative mouse Wnt pathway gene targets (Qiagen) was used to analyze RNA from mouse control and 5-day post-kainate hippocampi. The results were expressed as a volcano plot (left). Genes to the right or left of the vertical grey lines indicate >2-fold (+/-) changes in gene expression. Genes above the horizontal grey line indicate significant changes in the cohort. A list of Wnt-target genes that were significantly induced >2-fold is shown (right). (E, F) Wnt signaling occurs in neurons. Representative confocal images (x63) show the neuronal localization of pGSK3β$^{s9}$ (green), and (F) β-catenin (green) in hippocampus at day 5 post-SE with pilocarpine, when compared to control mice. [DAPI (blue; all nuclei), GFAP (red; astrocytes) in (E) and NeuN (red, neurons) in (F)]. The percentage of positive pGSK3β$^{S9}$ in (E), and of β-catenin in (F) neuronal cells of the dentate gyrus (DG) region at 5-days post-SE with pilocarpine and in control mice data are compiled from multiple sections (see Methods) and represented as mean ± SE. $^*P < 0.05$, $^{**}P < 0.01$, $^{***}P < 0.001$ by unpaired student's *t*-test, using Prism 9.0 (Graphpad) or MS-Excel. Refer to supplemental materials for more details on the number of replicates and statistical analyses for each experiment.

## Results

### Wnt signaling is activated during the epileptogenic period

The inhibition of glycogen synthase kinase 3 beta by serine 9 phosphorylation (pGSK3β$^{S9}$) is necessary for accumulation of β-catenin protein. The inhibitory GSK3β$^{S9}$ phosphorylation and β-catenin are consensus hallmarks of elevated Wnt signaling (reviewed in [39]). In a time-course analysis following induction of status epilepticus (SE) by pilocarpine (Fig 1B), there was a slight induction of pGSK3β$^{S9}$ by day 3, reached a maximum by 5 days and then declined to control levels by 7 days post-SE. Consistently, Fig 1B shows that β-catenin expression mirrored GSK3β phosphorylation, with a maximal increase at 5 days and a decline to control levels at 7 days. The changes in GSK3β phosphorylation and β-catenin levels appear to be general features of epileptogenesis, as SE induction by kainate, which induces SE through a different receptor-based mechanism, but nonetheless yielded similar results (S1A Fig). These data underscore that the changes in Wnt signaling are associated with SE and are independent of the mode of seizure induction. Finally, post-SE, Wnt signaling is also induced in rats following pilocarpine-induced SE, albeit with a slightly different time-course (S1B Fig) and additionally demonstrate a phenomenon that is independent of rodent species.

The gene expression pattern triggered by Wnt signaling is critical to the functions and phenotypes for different tissues. To expand the observations of Fig 1B, we used qRT-PCR experiments at 5 days post-SE to show that the gene expression of Wnt target gene Axin 2 (pilocarpine–Fig 1C, kainate–S1C Fig) was induced. Next, we used an unbiased mouse qRT-PCR-based Wnt pathway array to survey a wide range of known Wnt target genes induced by kainate at 5 days post-SE. As seen in Fig 1D, numerous known Wnt pathway target genes were induced at this 5-day early epileptogenic period after kainate treatment (as depicted by volcano plot analysis). The activation of Wnt signaling at day 5 post-SE appears to be a general feature of early epileptogenesis, regardless of the chemoconvulsant used for induction (kainate (for C57Bl6 mouse strains) or pilocarpine (for FVBN mouse strains), or of species (mouse or rat).

Activation of Wnt signaling can be achieved by induction of Wnt ligands, inhibition of Wnt repressors, or a combination of both mechanisms. We chose to examine three candidate Wnt inhibitor genes (HBP1, sFRP1, and FoxG1). We previously showed the HMG box protein 1 (HBP1) transcriptional repressor is an inhibitor of Wnt signaling and enforces senescence [29, 30]. In the brain, HBP1 additionally has a role for enforcing cortical differentiation, as demonstrated in an elegant mouse model [40], consistent with our other work in non-neuronal tissues [29, 30, 41–44]. We also examined other Wnt pathway inhibitors such as secreted Frizzled Related-Protein 1 (sFRP1), which inhibits Wnt pathway activation by extracellularly sequestering Wnt ligands from the FZD receptors [28]. The sFRP family and its epigenetic suppression

is a feature of many cancers and other diseases and highlights the importance of modulating Wnt signaling [27, 45]. Lastly, we selected Forkhead box protein G1 (FoxG1) because of previous reports that FoxG1 inhibited Wnt signaling in early neuronal development (reviewed in [46, 47]. In examining these select Wnt pathway inhibitors, we observed that the expression of FoxG1, sFRP1, and HBP1 all declined following pilocarpine-induced SE at 5 days post-SE, coincident with the peak of Wnt signaling (Fig 1C). Similar results were observed with FoxG1 and HBP1 in kainate-induced SE at 5 days post-SE (S1C Fig). Concomitant with the decline in Wnt pathway inhibitory proteins and with the increased activation of Wnt target genes, a Wnt ligand (Wnt8B) increased at 5 days post-SE in both pilocarpine-induced SE (Fig 1C) and kainate-induced SE (S1C Fig). Together, these results support the notion that Wnt signaling induction resulted from a combination of induction of at least one Wnt ligand (e.g., Wnt8B) and decreased expression of Wnt pathway inhibitors (sFRP1, HBP1, and Fox G1). Further, the similarity of Wnt activation mechanism suggests that Wnt pathway activation is a common response to SE in the hippocampus, regardless of how SE was experimentally triggered.

To determine where hippocampal Wnt signaling occurs, we examined the localization of pGSK3β$^{S9}$ at day 5 post-SE using confocal microscopy. A few pGSK3β$^{S9}$ positive neurons in the hilus of the dentate gyrus (DG) were observed in coronal sections from control mice, but showed little signal above background elsewhere (S1D Fig). In contrast, pGSK3β$^{S9}$ was strongly induced throughout neurons in the DG and CA1-3 (CA1; Fig 1E and DG; S1D Fig). pGSK3β$^{S9}$ was also observed in a smaller subset of astrocytes (less than 20%, Fig 1E). β-catenin expression in control and 5-day post-SE mice showed nearly identical results to pGSK3β$^{S9}$, with broad expression in neurons throughout the hippocampus (Fig 1F, S1E Fig). A similar broad induction of Wnt signaling was additionally observed with a transgenic Wnt reporter strain (C57Bl/6 background, for kainate experiments) at five days following kainate-induced SE (S1F Fig). Thus, Wnt signaling appears to be strongly induced 5 days after seizure induction, as shown by the biochemical and cellular analyses.

## Induction of Wnt signaling in the epileptogenic period results in aerobic glucose metabolism

We next sought to understand the consequences of Wnt signaling in this early period of epileptogenesis. Elegant studies demonstrated that constitutive Wnt signaling is required for the low-level proliferation and neurogenesis observed in the adult hippocampus SGZ [37, 48]. Consistently, we did observe a few cells in the hilus or subventricular zone (SVZ) that are strongly positive for pGSK3β$^{S9}$ (S1D Fig), but otherwise, there was little signal (Fig 1E and 1F). Others have described a well-documented program of increased, but relatively low-level proliferation in the hippocampus following SE [49–52]. In contrast to this low-level proliferation, we observed widespread Wnt signaling at 5 days post-SE (Fig 1E and 1F, S1D–S1F Fig). This level appeared inconsistent with the small-scale neuronal stem cell proliferation observed in multiple studies. Indeed, Wnt targets associated with proliferation such as Myc and cyclin D1 (from tumor cells; reviewed in [53]) were also not widely induced at 5 days post-SE. Thus, we considered other possible functions.

Wnt signaling in cancer is linked to a Warburg metabolic reprogramming in which tumor cells that have an altered metabolism that is characterized by increased glucose uptake and usage as a means to provide energy and biosynthetic precursors for sustained proliferation [24, 26]. We hypothesized that a similar metabolic shift might occur in the hippocampus during the epileptogenic period, but its role would not be through proliferation. If re-interpreted through the lens of Wnt signaling in cancer cells, some evidence seems to exist. One consequence of Wnt signaling is the inhibition of GSK3β. Rats that were fed the GSK3β inhibitor

LiCl demonstrated increased hippocampal glucose metabolism [54], while LiCl treatment of cultured neurons and astrocytes had increased glucose consumption, including lactate export, a signature manifestation of a Warburg effect, i.e., aerobic glycolysis with lactate production [38, 55].

We directly tested pyruvate metabolism, based on the following information. One of the Wnt target genes identified in a Wnt-driven metabolic re-programing/Warburg effect is pyruvate dehydrogenase kinase (PDK1 and PDK4, [24]). The PDK enzymes phosphorylate and negatively regulate pyruvate dehydrogenase (PDH), which serves as a gatekeeper for the Tricarboxylic acid (TCA) cycle [56]. If PDH is inhibited by phosphorylation, pyruvate is not converted into acetyl CoA for TCA cycle entry. Instead, when PDH is inhibited, the accumulated pyruvate will be converted into lactate, the signature metabolite of the Warburg effect. Thus, PDH restriction also limits metabolic flux through the TCA cycle, preventing mitochondrial damage due to excess oxidative phosphorylation and subsequent generation of reactive oxygen species [57]. PDK1 is expressed predominantly in astrocytes [58], and its expression declined 5 days post-SE (S2A Fig), so it is unlikely to be regulated by post-SE Wnt signaling (Fig 1). In contrast, PDK4 protein and gene expression increases post-SE (Fig 2A), consistent with observed Wnt activation (Fig 1B–1D) and mirroring the temporal progression of Wnt signaling (Fig 1E–1F). Furthermore, PDK4 was induced predominantly in neurons and in a small population of astrocytes, while peaking at day 5 post SE (Fig 2B). We next asked if the induction in PDK4 altered the phosphorylation and regulation of the crucial PDH enzyme at serine 293. The phosphorylation of Ser 293 results in the inhibition of PDH and of acetyl CoA production, effectively shunting the pyruvate carbons to lactate production. Consistent with PDK4 induction in this post-SE period, there was an increase in phosphorylation of PDH at serine 293 (pPDH$^{S293}$, Fig 2C). Lastly, PDH$^{S293}$ phosphorylation was induced predominately in neurons and a subset of astrocytes post-SE (Fig 2D), consistent with Wnt signaling and PDK4 induction. Thus, these results provided the first insight into a metabolic re-arrangement, orchestrated by the Wnt target PDK4 in neurons and astrocytes.

We next asked if there were additional changes in hippocampal metabolic enzymes with respect to a Warburg aerobic glycolysis. Concomitant with increased Wnt signaling at 5 days post-SE (Fig 1), we observed the transient induction of Warburg enzyme isoforms for hexokinase 2 (HKII), pyruvate kinase isoform M2 (PKM2) and lactate dehydrogenase isoform A (LDHA) at the mRNA and/or protein levels, for either pilocarpine (Fig 3A and 3B) or kainate-induced seizures (S3A and S3B Fig). The induction of Warburg metabolism (in cancer) commonly results in conversion of pyruvate to lactate via LDHA, but is usually contained within a single (tumor) cell type. Because the hippocampus has multiple cell types, we next examined the localization of the Warburg isoform of pyruvate kinase (PKM2) using confocal microscopy and found expression within a small fraction of hippocampal astrocytes in control mice. In contrast, at 5 days post-SE, a large fraction of hippocampal astrocytes expressed PKM2 (Fig 3C). Virtually no PKM2 was expressed in neurons at any time post-SE.

From the mRNA and protein expression of key enzymes and their modification state, the results of Figs 2 and 3 predict that a significant shift in metabolism occurs at 5-days post-SE, consistent with a reprogramming to Warburg (aerobic) glycolysis. But the localization studies reveal additional complexity in that astrocytes and neurons within the hippocampus comprise different enzymatic reactions of the apparent overall Warburg metabolism. In the early epileptogenic period, glycolytic PKM2 increased in the astrocytes, whereas the TCA gateway regulator PDK4 and PDH$^{S293}$ phosphorylation occurred mostly in neurons. These results are consistent with the proposed complex arrangement of overall metabolism between neuron and astrocyte, in which astrocytes export lactate for neuronal metabolic use [59], although it appears that stimulated neurons can also utilize glucose for metabolic purposes [60, 61]. Thus,

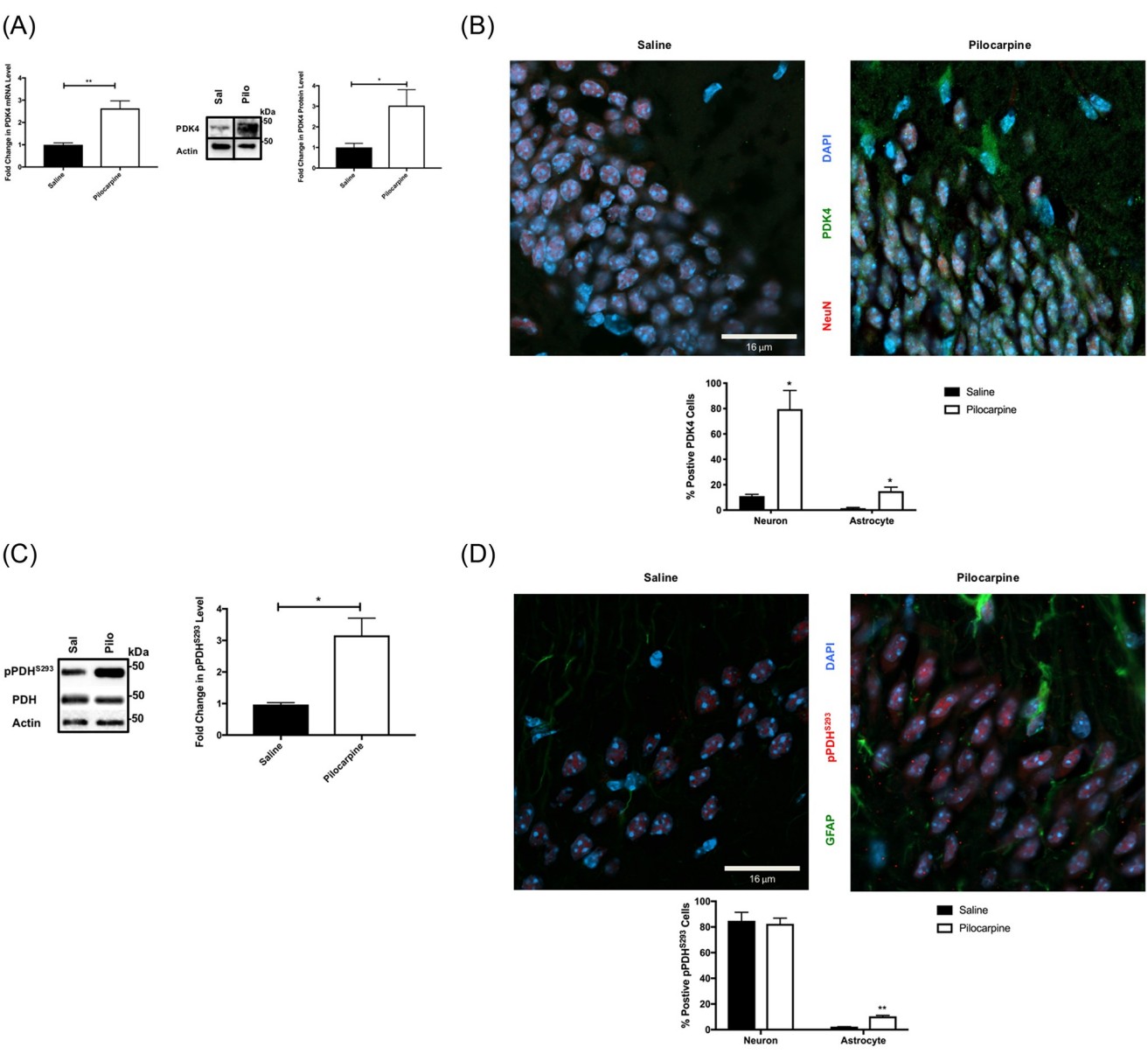

**Fig 2. Altered pyruvate metabolism at day 5 post-SE.** (A) Pyruvate dehydrogenase 4 (PDK4) mRNA and protein are increased at day 5 post-SE. Quantitative RT-PCR of PDK4 expression (left) and representative western blots with quantified data comparing the expression of PDK4 protein in hippocampus at day 5 following SE with pilocarpine (Pilo) or control (saline)–Right. (n = 4 mice/group). (B) PDK4 is localized primarily in neurons post-SE day 5. Representative confocal images (x63) and percentage of positive PDK4 (green) neuronal cells are higher in the DG region of 5 days post-SE with pilocarpine hippocampi when compared to control mice. [NeuN (red) and Dapi (blue)]. (C) Ser 293 phosphorylation of pyruvate dehydrogenase (PDH) is increased at day 5, post SE. Quantitative and representative western blot of pPDH$^{S293}$, total PDH, and actin (loading control) in hippocampi of day 5 induced SE with pilocarpine (Pilo) and control (Sal). (n = 4 for each group). (D) Confocal images (x63) and percentage of positive pPDH$^{S293}$ (red) neuronal cells (see supplemental methods) are higher in the DG region of 5 days post-SE with pilocarpine hippocampi when compared to control mice. [GFAP (green) and Dapi (blue)]. Data are represented as mean ± SEM. *$P < 0.05$, **$P < 0.01$ by unpaired student's *t*-test using Prism 9.0 (Graphpad). Refer to supplemental materials for more details on the number of replicates and statistical analyses for each experiment.

the Warburg metabolic re-arrangement in the hippocampus at day 5 post SE, while comprised of different metabolic enzymes, maintains the functional split between hippocampal neurons and astrocytes, implying a new and unappreciated flexibility in Warburg metabolic reprogramming in the hippocampus.

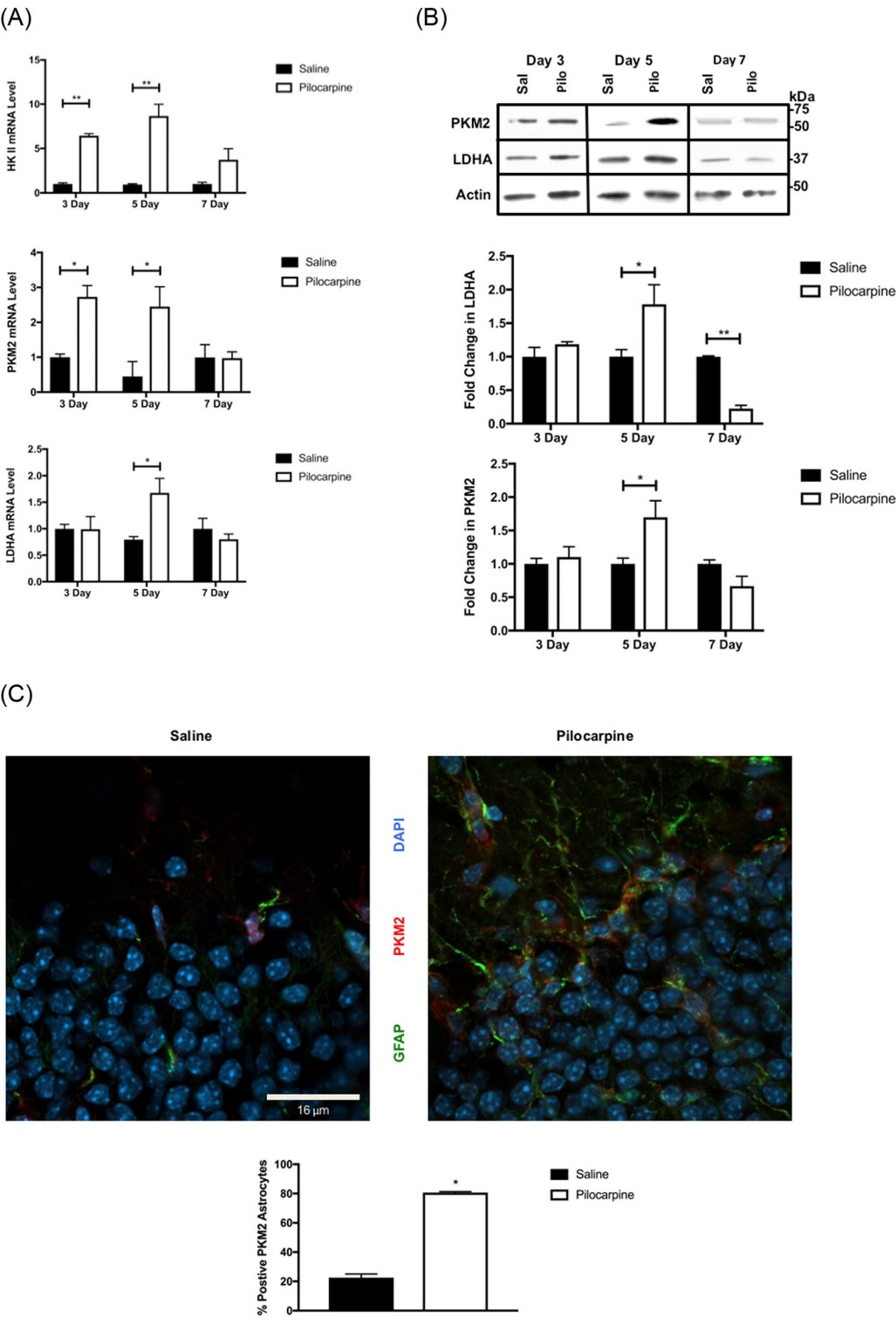

**Fig 3. Status-induced Warburg metabolic enzymes.** (A) The mRNA expression of Warburg metabolic enzyme isoforms is elevated after pilocarpine-induced SE. Quantitative RT-PCR was used to measure the mRNA for hexokinase 2 (HK II), pyruvate kinase M2 isoform (PKM2) and lactate dehydrogenase A-isoform (LDHA) from the hippocampi of the 3-, 5- and 7-day post-status epilepticus mice in (pilocarpine-treated) compared to control (saline) mice. (B) Protein expression of Warburg metabolic enzyme isoforms are elevated after pilocarpine-induced SE. Representative and quantified western blots of hippocampal extracts from 3-, 5- or 7-day pilocarpine (Pilo), or control saline (Sal) are shown for PKM2, LDHA,

and actin (loading control, n = 3–5 for each group). (C) Differential localization of PKM2 after pilocarpine-induced SE. Confocal images (x63) and percentage of positive PKM2 (red) astrocytes (see supplemental methods) are higher in the DG region of 5 days post-SE with pilocarpine hippocampus when compared to control mice. [GFAP (green) and Dapi (blue)]. Data are represented as mean ± SEM. *$P <0.05$, **$P< 0.01$ by unpaired student's $t$-test using Prism 9.0 (Graphpad). Refer to supplemental materials for more details on the number of replicates and statistical analyses for each experiment.

## Metabolomic analysis demonstrates a shift to Warburg-like metabolism in the epileptogenic period

The induction of Warburg glycolytic enzymes and alteration of PDH activity suggests a shift in metabolism, but any Warburg reprogramming should be evident in the metabolite patterns. Therefore, we measured steady-state metabolites from control and 5 days post-SE hippocampi by Nuclear Magnetic Resonance (NMR). A representative 1-D 1H NMR spectrum is shown in S4A Fig. The most striking result is a clear accumulation of lactate, but more detailed analysis was required to discern the pattern of metabolic pathways. The significant metabolite changes are represented in a hierarchical cluster dendrogram in Fig 4A (derived from S1 Table). Metabolite set enrichment analysis (MSEA) was performed using Metaboanalyst 4.0 [62] to identify pathways from the Human Metabolome Database that might be altered 5 days post-SE. Remarkably, this unbiased analysis (Fig 4B) indicated that Warburg metabolism is the most significant metabolic change, confirming the enzyme changes observed in Fig 3. Other metabolic pathways from Fig 4B, in particular changes in glutamate metabolism, are later discussed in context. Principle component analysis (PCA, Fig 4C) and partial least-squared analysis (PLSA, S4B and S4C Fig) of the measured hippocampal metabolites provided independent tests of statistical significance in the 5-day post-SE period.

Increased glucose flux often occurs with a Warburg re-arrangement, so flux measurements are often informative. To confirm the shift to aerobic glycolysis, we examined non-steady-state glucose metabolism, or glucose flux to follow the initial incorporation of a pulse of [U $^{13}$C]-glucose into the TCA cycle before multiple rounds had occurred (see Methods). As predicted by the steady-state analysis, [3-$^{13}$C]-lactate and [2-$^{13}$C]-lactate, derived from glucose via glycolysis to pyruvate, was strongly elevated in 5 days post-SE hippocampus (Fig 4D and 4E, see also S2 Table, S4D Fig). Another indicator of pyruvate status derived from transamination of pyruvate, [3-$^{13}$C]-alanine, was also elevated, although the increase in steady-state alanine was not significant, suggesting the flow from glucose into alanine was transient. These results are consistent with the observed increase in steady-state levels of lactate and thus support the conclusion of increased aerobic glycolysis, i.e., a Warburg metabolic re-arrangement.

## Altered glutamine-glutamate-GABA metabolism in the epileptogenic period

The second most significant metabolic pathway altered in the unbiased MESA analysis of 5-day post-SE metabolism was glutamate (Fig 4B). Glutamate plays a central role in brain metabolism, functioning as an excitatory neurotransmitter, an energy source, a nitrogen source and the metabolic precursor for the inhibitory neurotransmitter GABA (reviewed in [64]). Further, glutamate is recycled from the synapse by astrocyte uptake, converted to glutamine, transported back to neurons where it is reconverted to glutamate (Fig 4E). Alterations to glutamine-glutamate-GABA homeostasis have been observed in chronic seizures [65], and many pharmacological interventions in epilepsy are based upon restoring GABA levels. Because PDH controls the flux into the TCA cycle and thus the synthesis of glutamate from 2-oxoglutarate, we probed whether the Warburg metabolic rearrangements in epileptogenesis, or other changes in enzymes impact glutamine-glutamate-GABA homeostasis.

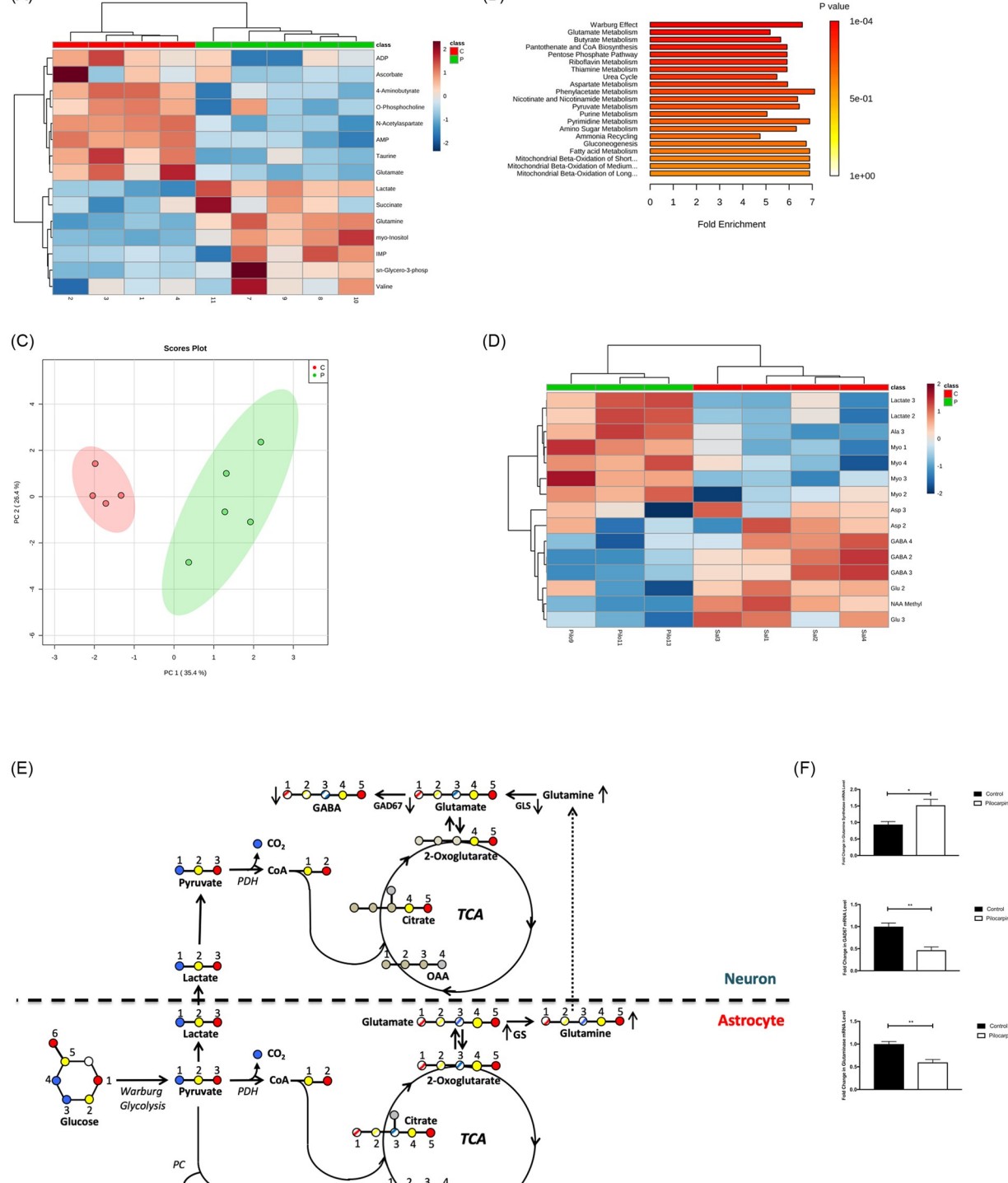

**Fig 4. Metabolite analysis reveals elevated lactate and glucose metabolism in 5-day post-SE hippocampal metabolism—Signatures of Warburg metabolism.** 4–5 different hippocampi per group were independently processed and analyzed as described in methods. (A) Hierarchical cluster analysis of metabolites consistent with Warburg metabolism (Metaboanalyst 4.0, [62, 63]). The top 15 metabolites by significance are displayed. Metabolites were segregated into two clusters with clear up- and down-changes in metabolites between control and pilocarpine-treated mice. (B) Warburg effect identified by metabolite set enrichment analysis (MSEA, Metaboanalyst 4.0, [62, 63]). The top 20 metabolic pathways enriched in 5-day post-SE hippocampus relative to control were identified by analysis of metabolites from S1 Table using

MSEA. *P*-value and fold enrichment are displayed. (C) Principal component analysis (Metaboanalyst 4.0, [62, 63]) using the metabolites from S1 Table demonstrates a significant difference between control and 5-day post-SE metabolism. The metabolite principal components showed significant separation (>95% confidence) between control mice (red) and 5-day post pilocarpine-treated mice (green) in 2D plot. (D) Hierarchical cluster analysis of $^{13}$C metabolite carbons demonstrates glucose flux into lactate (Metaboanalyst 4.0, [62, 63]). The top 15 $^{13}$C-labeled metabolites by significance are displayed. Unbiased analysis of $^{13}$C-labeled metabolites segregated into two clusters with clear up- and down-changes in metabolites between control and 5-day post-pilocarpine-treated mice. (E) Schematic representation of possible isotopmers arising from [U-$^{13}$C] glucose in glycolysis in the first turn of the TCA cycle. Each glucose carbon is color-coded for its fate in glycolysis, with the glucose split at the aldolase step, eventually ending at pyruvate. Astrocyte pyruvate is converted to lactate, transported to neurons and subsequently re-converted to pyruvate, where it enters the TCA cycle via pyruvate dehydrogenase (PDH). This gives labeling citrate, 2-oxoglutarate, glutamate and GABA on carbons 4 and 5. Pyruvate remaining in astrocytes can similarly be converted via PDH, but can also undergo the anaplerotic pyruvate carboxylase step, converting to oxaloacetate (OAA) with label at carbons 1, 2 and 3. Subsequent condensation of OAA with acetyl CoA yields some of the citrate pool labeled at carbons 1–5, readily detectable by NMR via the interaction of carbon 3 with carbon 4. GAD67, Glutamate decarboxylase; GS, Glutamine synthetase; GLS, Glutaminase. (F) GAD67, GS, and GLS mRNA Levels at day 5 post-SE. GAD67 and Glutaminase mRNA levels decreased while glutamine synthetase mRNA increased in hippocampus at 5 days post-SE with pilocarpine. (n = 4–6 mice per group). Data are represented as mean ± SEM. *$P$ <0.05, **$P$ < 0.01 by unpaired student's *t*-test using Prism 9.0 (Graphpad). Refer to supplemental materials for more details on the number of replicates and statistical analyses for each experiment.

With metabolism split between astrocytes and neurons, and with PDH phosphorylation observed mainly in neurons (Fig 2C and 2D), we reasoned that a primary function for a War-burg/aerobic glucose metabolism could impact glutamate/GABA homeostasis within the hippocampus 5-days post-SE. The decline in steady-state glutamate (Fig 4A, S1 Table) was consistent with the decrease in glucose labeling of glutamate via 2-oxoglutarate in carbons 2 and 3, but not carbon 4 (Fig 4D, S2 Table–Glu2, Glu3, Glu4). The differential labeling provides an important clue into the 5-day post-SE alteration of glucose carbon incorporation into glutamate. As illustrated in Fig 4E, glucose metabolized in astrocytes via the astrocyte-specific, anaplerotic pyruvate carboxylase (PC) reaction results in labeling at carbons 1–3 in glutamate. Glutamate at carbons 4 and 5 are incorporated from pyruvate via PDH (Fig 4E). The observation that [U $^{13}$C]-glucose incorporation at carbon 4 does not change in epilepto-genesis suggests that PDH inhibition by phosphorylation is indeed required to maintain steady TCA activity under elevated glucose metabolism conditions. Decreased labeling of glutamate at carbons 2 and 3 was surprising and we first considered a decline in PC enzyme. However, there is no alteration of PC expression at day 5 in epileptogenesis (S4E Fig), sug-gesting other mechanisms.

A second pathway into glutamate carbon 4 may derive from the unidirectional glutamate-glutamine conversion and transport function of astrocytes (Fig 4E). Analysis of enzyme expression shows increased glutamine synthase (GS, astrocytic) but decreased glutaminase (GLS, neuron, Fig 4F), suggesting a potential decline in converting astrocyte-derived gluta-mate/glutamine back to glutamate in neurons. This was supported by examining steady-state glutamine levels, which were dramatically elevated at 5-day post-SE (Fig 4A, S1 Table), but with no change in glucose flux into glutamine (carbons 3 and 4, S2 Table). These results dem-onstrate the significance of reduced GLS enzyme and the correspondingly decreased glutamine reconversion back to glutamate.

Finally, the other metabolite in homeostasis with glutamate is GABA (4-aminobutyrate), which shows a decline of 20% in steady-state levels (Fig 4A, S1 Table). The decline in steady-state and glucose flux into GABA is nearly twice that of the decline in glutamate (S1 and S2 Tables), suggesting again that additional mechanisms besides PDH phosphorylation are in play. Indeed, expression of glutamate dehydrogenase (GAD67) mRNA, the key enzyme for GABA synthesis from glutamate, decreased at day 5 post-SE (Fig 4F), consistent with the decrease in GABA (Fig 4A, S1 Table) and published observations [66, 67]. The alterations observed in metabolites, [U-$^{13}$C]-glucose labeling and enzyme expression within the neuron-astrocyte homeostatic metabolism are summarized in Fig 4E. These data further inform the extent of metabolic reprogramming in the hippocampus that is depicted in Fig 5F. Lastly,

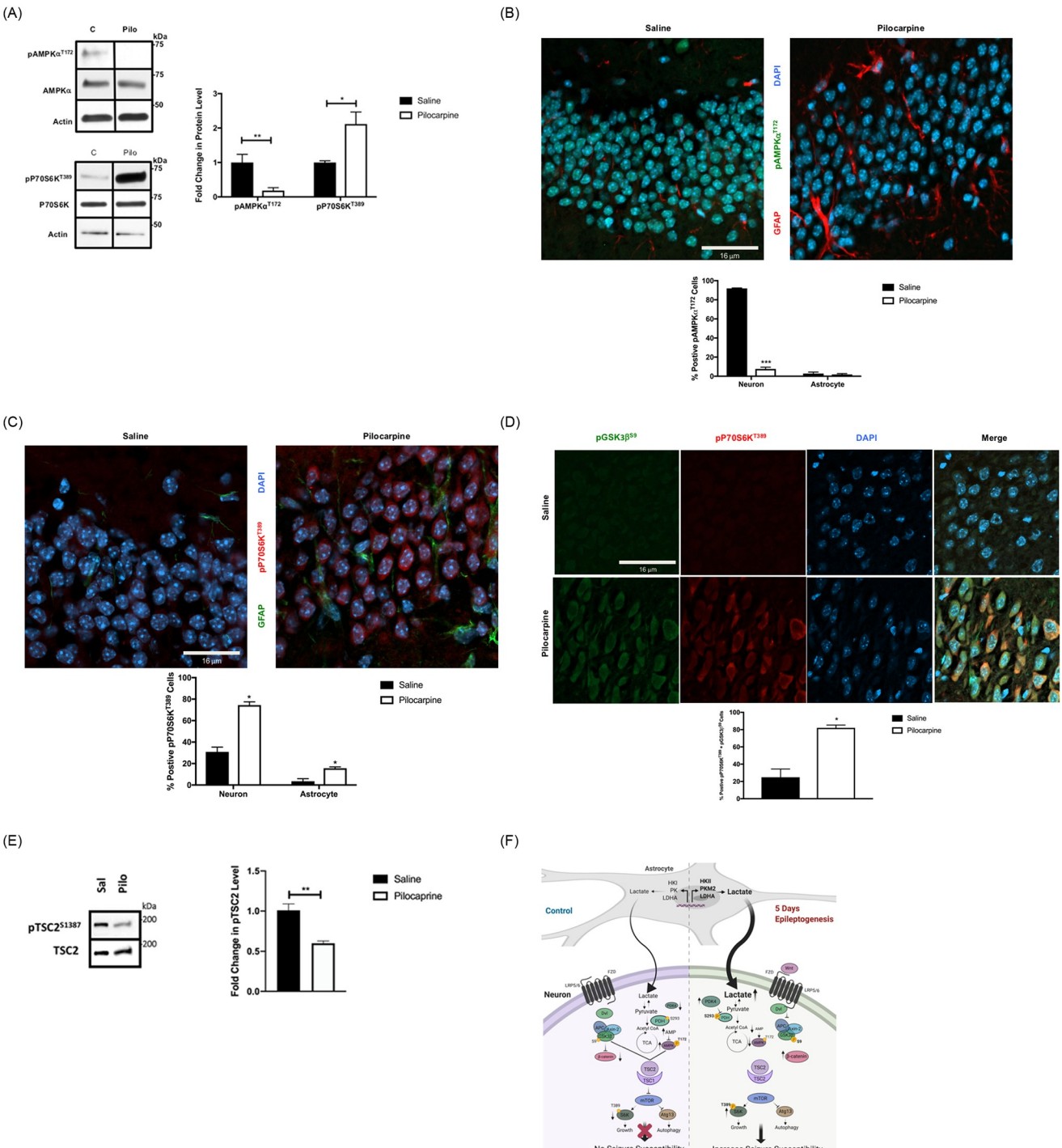

**Fig 5. Induced Wnt signaling post-SE coordinates AMPK and mTOR signaling pathways.** (A) Coordinated inhibition of AMPK and activation of P70S6 Kinase. (Left) Representative western blots with quantification (Right) comparing the expression of pAMPKα$^{T172}$ and pP70S6K$^{T389}$ in hippocampi at day 5 following SE with pilocarpine (Pilo) or control (saline). (n = 3–4 mice/group). Data are represented as mean ± SEM. $^*P < 0.05$, $^{**}P < 0.01$ by unpaired student's $t$-test using Prism 9.0 (Graphpad). (B and C) Localization of AMPK and mTOR activation by confocal microscopy. Representative confocal sections with quantified analysis of positive pAMPKα$^{T172}$ (green; B) and pP70S6K$^{T389}$ (red; C) neuronal cells (see supplemental methods) are increased in the DG region of 5-day post-pilocarpine induced-SE when compared to control saline. [GFAP (red in B; green in C) and Dapi (blue)]. Data are represented as mean ± SEM. $^*P < 0.05$, $^{***}P < 0.001$ by unpaired student's $t$-test using Prism 9.0 (Graphpad). (D) pP70S6K$^{T389}$ and pGSK3β$^{S9}$ co-localize to neurons. Immunostaining with pP70S6K$^{T389}$ (red) and pGSK3β$^{S9}$ (green) show both co-localize to neuronal cells and are higher in percentage (see supplemental methods) at day 5 post-SE with pilocarpine when compared to control. [Dapi (blue)]. (E) TSC2 phosphorylation inhibited at day 5 post-SE. Representative

and quantified western blots of hippocampal extracts from pilocarpine- treated (Pilo), or control saline (Sal) are shown for pTSC2$^{S1387}$, and total TSC2 (loading control). (n = 3–5 for each group). Data are represented as mean ± SEM. $^{**}P < 0.01$ by unpaired student's $t$-test using Prism 9.0 (Graphpad). Refer to supplemental materials for more details on the number of replicates and statistical analyses for each experiment. (F) Summary and mechanistic model of Wnt-AMPK-mTOR signaling in epileptogenesis. Activation of canonical Wnt signaling results in inhibition of GSK3β, increases β-catenin protein levels, resulting metabolic gene expression, including increases HKII, PKM2 and LDHA consistent with activation of aerobic glucose metabolism. This "Warburg" metabolism results in decreased AMP and inhibition of AMP-dependent protein kinase (AMPK). The coordinated inhibition of GSK3β and AMPK, TSC2 from its inhibitory role in the mTOR pathway, resulting in mTOR activation and a change in seizure susceptibility. The figure was created with BioRender.com.

these data suggest that a net effect of the metabolic rearrangements is the well-known and signature decrease in GABA in epileptogenesis [67, 68]. Together, these diverse data link glucose usage and the flow of carbons into GABA-glutamate-glutamine metabolism at day 5 post-SE into the signature changes in GABA during epileptogenesis. Finally, our data demonstrate that the split Warburg rearrangement also links glucose carbon flow to the altered glutamate/GABA ratio in epileptogenesis.

## Wnt signaling and aerobic glucose metabolism coordinate AMPK-mTOR signaling pathways in the epileptogenic period

The metabolomics and altered glycolysis/TCA enzyme expression (Figs 3 and 4) additionally suggested a link into mTOR signaling associated with epileptogenesis [4, 21], possibly through AMPK regulation of TSC2. The increased glucose flux and decrease in AMP at 5 days post-SE (Fig 4A, S1 Table, and S4A Fig) suggested that AMP-dependent protein kinase (AMPK) activity should decrease. We further hypothesized that the decrease in AMP would initiate a signaling cascade resulting in mTOR activation in collaboration with GSK. Mechanistic analyses from other fields indicate that both Wnt signaling and AMPK activity may dictate mTOR activation (Fig 1A). Thus, this section explores the signaling outputs at day 5 of epileptogenesis by measuring the phosphorylation of AMPK and TSC2—two related inputs that regulate mTOR activation ([21], Figs 1A and 5E).

In response to increases in AMP concentration, Liver Kinase B1 (LKB1) phosphorylates AMPKα at threonine 172 only if AMP is bound to AMPKγ [69, 70], resulting in activation of AMPK. As shown in Fig 5A, we found that pAMPKα$^{T172}$ was high in control hippocampus, consistent with a high AMP level, then declined at 5 days post-SE, consistent with decreased AMP levels (Fig 4A, and S1 Table). Confocal microscopy localization of pAMPKα$^{T172}$ was entirely neuronal in control hippocampus, with a sharp decline 5 days post-SE (Fig 5B), confirming the protein analysis of Fig 5A. Furthermore, because the changes in pAMPKα$^{T172}$ are specifically neuronal, we conclude energetically that the TCA-cycle metabolic transitions and concomitant AMP levels measured by NMR metabolomics are likely neuronal, and consistent with changes in glucose flux triggered by Wnt signaling.

AMPK is a major coordinator of metabolic transitions, including mTOR activation. The phosphorylation of TSC2 by AMPK, and secondarily by GSK3β following AMPK priming, promotes the dissociation of TSC1 and TSC2 and de-inhibits mTOR (Fig 5F, reviewed in [71]). The results of Figs 1 and 5A and 5B, in which GSK3β and AMPK are coordinately inhibited post-SE, suggest that there should be a concomitant increase in mTOR signaling, which is an iconic event in epileptogenesis and epilepsies [4]. mTOR activation was measured by increased phosphorylation of the downstream mTOR effector P70S6K at threonine 389 (pP70S6K$^{T389}$, Fig 5A), and consistent with published work [4]. Induction of pP70S6K$^{T389}$ appears to be mostly neuronal, as determined by confocal microscopy (Fig 5C) and consistent with the predominantly neuronal co-localization of changes in pAMPK$^{T172}$ and pGSK3β$^{S9}$. A key intermediate in the activation of mTOR is the AMPK-dependent phosphorylation of

TSC2. There is genetic and biochemical evidence that inactivation of TSC2 is important for mTOR activation. For example, TSC deletions are associated with seizures in human and in animal models [5, 6, 13]. We next measured the phosphorylation of serine 1387 of TSC as a functional output of AMPK activity. A decrease TSC$^{S1387}$ phosphorylation would be associated with heighted mTOR activation, as detected by pP70S6K$^{T389}$. As shown in Fig 5E, phosphorylation of TSC2 at the AMPK site (pTSC$^{S1387}$, measured by western blot) as a ratio of phosphorylated to total TSC2 was significantly decreased at day-5 post-pilocarpine.

Fig 5F summarizes the findings from Figs 1–5, highlighting that the coordinated induction of Wnt signaling and transition to Warburg-like metabolism orchestrated jointly by astrocytes and neurons appears to regulate events that are known features of epileptogenesis, including neurotransmitter ratios and mTOR activation. Wnt signaling has two apparent functions—1) inhibition of GSK3β and 2) PDK4 and regulation of PDH. The metabolic consequence in Fig 4 further details the impact of glutamine-glutamate-GABA homeostasis in regulating the important ratio of glutamate/GABA in early epileptogenesis. In Fig 5, we also detailed the signaling events leading to metabolic rearrangement and to mTOR activation or disinhibition which features an intricate mechanism involving AMPK, and GSK3β, TSC2 in the early epileptogenic period following SE. The Wnt signaling-induced decrease in GSK3β and AMPK activities then coordinate activation of the mTOR pathway. As an initial test of this model, pGSK3β$^{S9}$ and pP70S6K$^{T389}$ co-localized in the same cells (Fig 5D). Thus, Fig 5F summarizes the data to this point with a useful signaling and metabolic context. Figs 6 and 7 both build upon and test the model defined in Fig 5F to solidify the reaction order within the model.

## Epileptogenic metabolic reprogramming is situated between Wnt signaling and AMPK and mTOR signaling

The experimental framework of Fig 5F predicts that the interlinked Wnt signaling and Warburg metabolism of the epileptogenic period drives the intricate signaling changes leading to mTOR activation. Because TSC2 integrates the signaling input of GSK3β and AMPK [21], we investigated the consequences of blocking the split Warburg effect using the glycolytic inhibitor 2-deoxyglucose (2DG) in the early epileptogenic period, altering metabolic input from AMPK and thus mTOR signaling. 2DG is a potent inhibitor of glucose metabolism and has been used in some pre-clinical models of epilepsy, supporting the notion that aberrant glucose metabolism may constitute a contributing etiology [74, 75]. Experimentally, we quantitated how 2DG influenced Wnt-, AMPK- and/or mTOR signaling by the criteria set forth in Figs 1–5. Mice were treated with 2DG (see Methods, [74]) for 5 days post-SE. We found no change in Wnt activation as determined by pGSK3β$^{S9}$ levels (Fig 6A), nor any effect on post-SE Wnt target gene PDK4 induction or its substrate PDHα$^{S293}$ phosphorylation (Fig 6A and 6B, S5A Fig). Further, the post-SE induction of PKM2 was not affected by treatment with 2DG, confirming that PKM2 in astrocytes was not induced as a result of glucose influx (Fig 6A). However, treatment with 2DG affected both AMPK and mTOR signaling. As shown in Fig 6C and 6D, the decrease in pAMPKα$^{T172}$ at 5 days post-SE was completely reversed by treatment with 2DG, demonstrating the efficacy of inhibiting glycolysis on post-SE energy state (Fig 6C and 6D). As expected, control mice treated with 2DG showed a small but insignificant increase in pAMPKα$^{T172}$ compared to untreated controls. As predicted from the known signaling mechanisms (Fig 5F, [4, 21]), we observe that the induction of pP70S6K$^{T389}$ 5 days post-SE is ablated by 2DG treatment (Fig 6C and 6E). Therefore, both the levels and the localization of pAMPKα$^{T172}$ and pP70S6K$^{T389}$ following treatment with 2DG is restored to levels in the saline-controls.

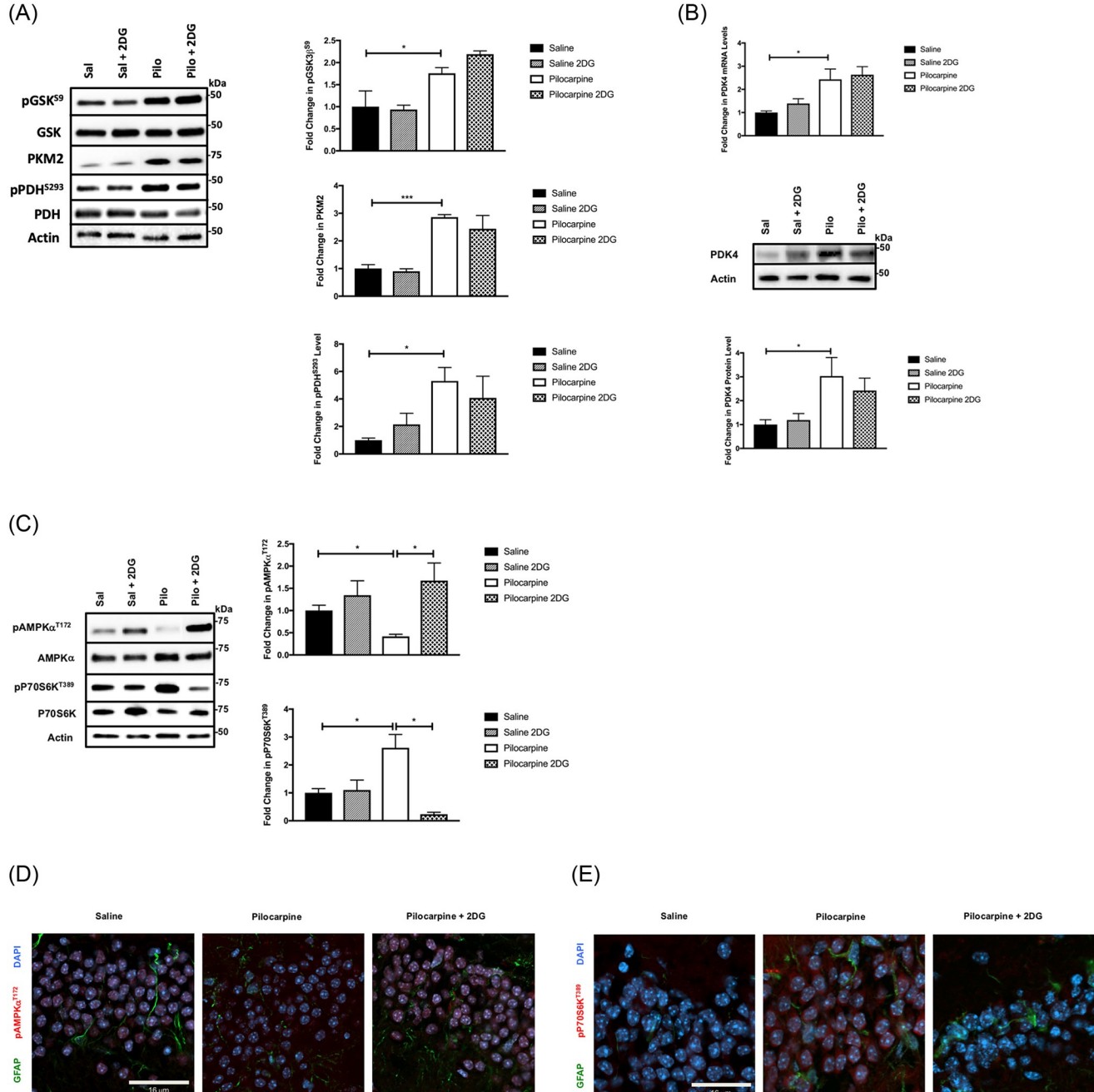

**Fig 6. Glycolytic inhibitor 2-DG alters downstream AMPK and mTOR signaling, and neuronal gene expression, but does not affect Wnt signaling.**
(A) SE-induced Wnt signaling and Warburg metabolic enzyme PKM2 are unaffected by 2DG. The protein expression of pGSK3β$^{S9}$, pPDH$^{S293}$, and PKM2 in the epileptogenic period was measured by western blot. Representative blots are on the left and quantified data (n = 4–6 mice per group) on the right. Data are represented as mean ± SEM. *$P$ <0.05, ***$P$ <0.001 by unpaired student's $t$-test using Prism 9.0 (Graphpad). (B) The Wnt target gene PDK4 expression was unaffected by 2DG. Quantitative RT-PCR (top panel) shows PDK4 mRNA levels induced at 5-day post-SE relative to control. Representative western blots of PDK4 protein (center panel) and quantified data (bottom panel) are shown. There was no change in the expression levels of PDK4 in hippocampus at day 5 following SE with pilocarpine (Pilo) and treated with 2DG–right panel. (n = 4–6 mice per group). Data are represented as mean ± SEM. *$P$ <0.05, by unpaired student's $t$-test using Prism 9.0 (Graphpad). (C) SE-induced activation of AMPK (pAMPKα$^{T172}$) and of mTOR (pP70S6K$^{T389}$) is reversed with 2DG treatment. The levels of activated AMPK and activated P70S6K were detected by quantitative western blot from extract of the hippocampus from 5 days post SE mice, in the presences and absence of 2DG (see Methods). Data are represented as mean ± SEM. *$P$ <0.05 by unpaired student's $t$-test; (n = 4–6 mice for each group). Representative western blots with quantified data showed 2-DG treatment at day 5 post-SE with pilocarpine inhibit mTOR signaling downstream activation by decreased pP70S6K$^{T389}$ and activating pAMPKα$^{T172}$. (n = 4–6 mice for each group) Data are represented as mean ± SEM. *$P$ <0.05 by unpaired student's $t$-test using Prism 9.0 (Graphpad). (D and E) SE-induced localization of activated AMPK and mTOR activation is restored with 2DG treatment. Sections from 5-day post-SE hippocampus in the absence or presence of 2DG treatment (see Methods).

Confocal images (x63) from treated and control hippocampi were analyzed for positive pAMPKα$^{T172}$ (D, red), and pP70S6K$^{T389}$ (E, red) in the DG region. [GFAP (green) and Dapi (blue)]. Refer to supplemental materials for more details on the number of replicates and statistical analyses for each experiment.

The action of 2DG indicates that the split Warburg rearrangement is necessary for mTOR activation. Furthermore, the metabolic changes inhibited by 2DG are situated between Wnt signaling and mTOR activation. These results support the framework in Fig 5F for the 5-day post-SE period. Interrupting the high-glucose consuming Warburg metabolism with 2DG resulted in a reversal of 1) the epileptogenic decrease in AMPK activation; 2) the epileptogenic activation of mTOR; while 3) Wnt signaling and genes associated with establishing the Warburg effect are unaffected. As stated, 2DG is a potent inhibitor of glycolysis and glucose metabolism and of the subsequent energy state of the hippocampus. Because AMPK and GSK3β form an important regulatory nexus into the mTOR pathway (reviewed in [71]), sustaining AMPK activity at 5 days post-SE with 2DG treatment would be predicted to inhibit mTOR activation. Importantly, 2DG exhibited specificity in the framework of Fig 5F. 2DG had a clear non-effect on Wnt signaling, PDK4 expression and PDH activity, as evidenced by a lack of effect on pPDHα$^{S293}$ levels. Thus, 2DG uncoupled Wnt signaling from AMPK and mTOR signaling, squarely placing the split Warburg metabolism between Wnt and AMPK/mTOR networks.

## Constitutive Wnt signaling in a genetic model results in hypersensitivity to seizure induction with elevated mTOR downstream signaling

Because 2-DG treatment appeared to uncouple the transient Wnt signaling and metabolism/mTOR activation in the epileptogenic period, we considered and attempted a pharmacological approach to determine if blocking Wnt signaling also blocked downstream metabolic and AMPK/mTOR signaling. However, both LGK974 (a PORCN inhibitor) and XAV939 (a tankyrase inhibitor) failed to affect basal or induced hippocampal Wnt signaling. A likely reason is that these compounds did not penetrate the BBB. Clinical trials in cancers, especially for LGK974, have not focused on brain metastases ([76]; personal communications Dr. M. Maclaughlin, Dr. P. Brastianos). Thus, a direct pharmacological approach was not likely to be successful. We considered an intracerebral approach, but this was not likely to be physiological for future translational work.

As an alternative, we used a genetic approach with a mouse model of activated Wnt signaling. The specific investigation was to compare the molecular and metabolic mechanisms for epileptogenesis delineated in Figs 1–6 with a model in which Wnt signaling was triggered by genetic means and address whether these features could be general to Wnt signaling in the brain. A driving force was a convergent and unpublished project in our lab in which a mouse model with constitutive Wnt signaling (deletion of the HBP1 transcriptional repressor) exhibited spontaneous seizures and heightened sensitivity to seizures (see below). Our previous work had demonstrated that the transcriptional repressor HBP1 gene was an inhibitor of Wnt signaling by binding and inhibiting the LEF/TCF transcriptional activators. Decreases in HBP1 or an RNAi mediated HBP1 knockdown resulted in increased Wnt signaling, as would be expected for HBP1 functioning as a suppressor of Wnt signaling [29, 30, 77]. We then created a germline deletion of the HBP1 gene from ES cells containing a β-galactosidase gene trap in the HBP1 gene and then created pure-bred mice in the C57BL6 and FVBN backgrounds. To our surprise, spontaneous seizures in HBP1$^{-/-}$ mice were initially observed by our animal facilities and subsequently confirmed ourselves. This initially unexpected and puzzling phenotype prompted a closer examination when our studies of Wnt signaling in epileptogenesis

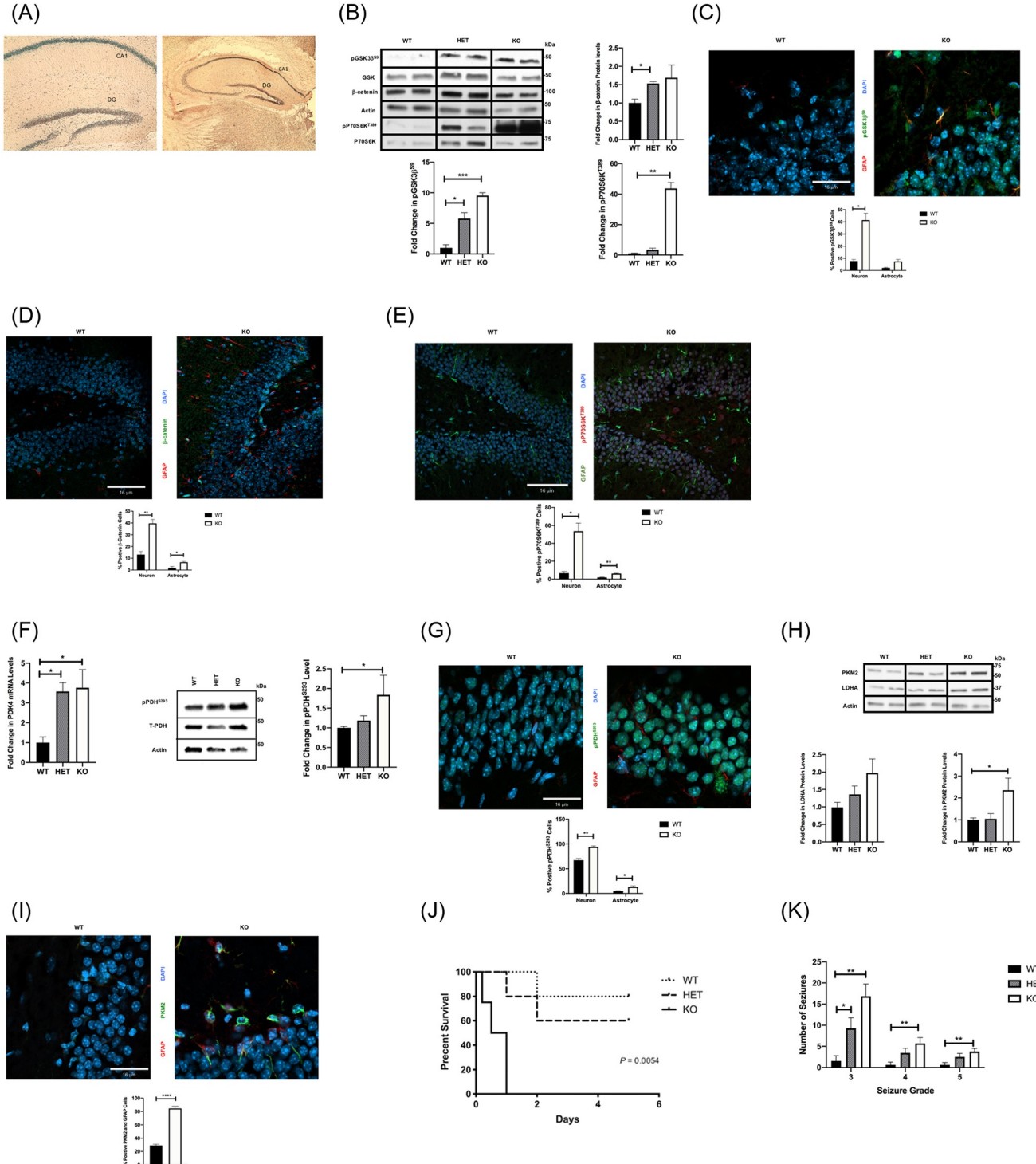

**Fig 7. The role of HBP1 and Wnt signaling in acquired seizure susceptibility.** (A) HBP1 gene is expressed in the CA1 and DG regions. The expression pattern of the HBP1 gene was determined in the HBP1$^{-/-}$ mouse through the inserted β-galactosidase gene. Note that the HBP1$^{-/-}$ mouse was generated by insertion of the β-galactosidase gene into intron 1 of the HBP1 genomic loci [72]. Thus, β-galactosidase activity is driven by the HBP1 promoter and is a reporter of the HBP1 promoter activity. Representative slides of HBP1$^{-/-}$ adult mouse hippocampus with β-galactosidase gene inserted into intron 1 of HBP1 gene. (B) Activation of Wnt and mTOR signaling in HBP1$^{-/-}$ and HBP1$^{+/-}$ mice. Representative western blot with quantified data of baseline levels of Wnt and mTOR signaling markers in FVBN mouse strain were increased in HBP1$^{-/-}$ knockout (KO) and HBP1$^{+/-}$ heterozygous (HET) when compared to control (WT). (C-E) Localization of activated Wnt and mTOR signaling in HBP$^{-/-}$ mice by confocal microscopy. Representative confocal sections with quantified analysis of positive pGSK3β$^{S9}$ (green; x63 in C), β-catenin (green; x20 in D), and pP70S6K$^{T389}$ (red; x20 in E) neuronal cells are increased in the

DG region of KO when compared to WT. [GFAP (red in C and D; green in E) and Dapi (blue)]. (n = 3–5 mice for each group). Data are represented as mean ± SEM. $^*P < 0.05$, $^{**}P < 0.01$, $^{***}P < 0.001$ by unpaired student's $t$-test using Prism 9.0 (Graphpad). (F) Wnt targets PDK4 and phosphorylation of pyruvate dehydrogenase (PDH) at Ser293 are increased in HBP1$^{-/-}$ mice. Quantitative RT-PCR of PDK4 gene expression (left) and representative western blot with quantified data (right) of pPDH$^{S293}$, total PDH, and actin (loading control) in hippocampi of HBP1$^{-/-}$ Knockout (KO), HBP1$^{+/-}$ heterozygous (HET), and control (WT) (n = 4 for each group). Data are represented as mean ± SEM. $^*P < 0.05$ unpaired student's $t$-test using Prism 9.0 (Graphpad). (G) Phosphorylated PDH$^{S293}$ localized in neuronal cells of HBP1$^{-/-}$ mice. Confocal images (x63) and percentage of positive pPDH$^{S293}$ (green) neuronal cells are higher in the DG region of KO when compared to WT mice. [GFAP (red) and Dapi (blue)]. Data are represented as mean ± SEM. $^*P < 0.05$, $^{**}P < 0.01$ by unpaired student's $t$-test using Prism 9.0 (Graphpad). (H) Elevation of Warburg metabolic enzyme isoforms in HBP1$^{-/-}$ mice. Quantitative and representative western blots of PKM2 protein levels were increased in KO group. (n = 3–5 mice for each group). Data are represented as mean ± SEM. $^*P < 0.05$ by unpaired student's $t$-test. (I) Localization of Warburg metabolic enzyme isoform (PKM2) in HBP1$^{-/-}$ mice. Confocal images (x63) and percentage of positive PKM2 (red) astrocytes (see supplemental methods) are higher in the DG region of 5 days post-SE with pilocarpine hippocampus when compared to control mice. [GFAP (green) and Dapi (blue)]. Data are represented as mean ± SEM. $^{****}P < 0.0001$ by unpaired student's $t$-test using Prism 9.0 (Graphpad). (J) Decreased survival after seizure induction in HBP1$^{-/-}$ KO mice. Survival curves were made for the first 5 days after seizure induction. (n = 5 mice for each group). Data are represented as mean ± SEM. $^*P < 0.05$ by unpaired student's $t$-test using Prism 9.0 (Graphpad). (K) Increased seizure sensitivity in HBP1$^{-/-}$ KO mice for FVBN strain. Results based on the number and severity of seizures on a 5-point modified Racine scale [73]. (n = 11–17 mice for each group). Data are represented as mean ± SEM. $^*P < 0.05$, $^{**}P < 0.01$ by unpaired student's $t$-test using Prism 9.0 (Graphpad). Refer to supplemental materials for more details on the number of replicates and statistical analyses for each experiment.

became clearer. In fact, Fig 1C of this paper showed that HBP1 levels were decreased in the 5-day period of early epileptogenesis (Fig 1C, [29, 30, 77]) and was coincident with a peak of Wnt signaling.

The configuration of the HBP1$^{-/-}$ mouse provided a useful tool, as the HBP1 gene was deleted in intron 1 in the germline with a β-galactosidase gene trap [72]. We could then use β-galactosidase as a proxy marker for HBP1 mRNA expressing cells. As shown in Fig 7A, HBP1 was expressed in both CA1 and DG, including the hilus, suggesting the distribution of HBP1 was plausible in regulating Wnt signaling in epileptogenesis (S1F Fig) and that HBP1 knockout mice offered a genetic means of testing the role of HBP1 and Wnt signaling in epileptogenesis. Thus, we posed two questions: 1) are the metabolic and signaling mechanisms from the day 5 pilocarpine and kainate models also utilized in a potential HBP1 genetic model of epilepsy? 2) Does HBP1 and Wnt signaling affect seizure susceptibility or set an epileptogenic threshold?

Comparing HBP1$^{-/-}$ and HBP1$^{+/-}$ mice with HBP1$^{+/+}$ mice thus provided an opportunity to test the model of Fig 5F. If genetic deletion of the HBP1 repressor increased Wnt signaling, then we could assess the consequences on metabolism and ensuing mTOR signaling. Using the approaches in Figs 1–6, we compared HBP1 gene dosage (HBP1$^{+/+}$, HBP1$^{+/-}$ and HBP1$^{-/-}$ mice) and Wnt signaling, predicting highest Wnt signaling in the full knockout. As shown in Fig 7B (FVBN background) and S6A Fig (C57BL/6 background), basal β-catenin and pGSK$^{S9}$ were elevated in HBP1$^{+/-}$ mice, and additionally so in HBP1$^{-/-}$ mice relative to HBP1$^{+/+}$ controls. Localization of Wnt signaling in the hippocampus as determined by confocal imaging of pGSK$^{S9}$ was predominantly neuronal, but with a small percentage of astrocytes (Fig 7C). Similar results were observed for β-catenin localization (Fig 7D). These results suggest that the predicted phenotype of constitutive Wnt signaling in the hippocampus was observed by knocking out the Wnt inhibitor HBP1.

We next probed whether constitutive Wnt signaling in HBP1 knockout mice phenocopied the other molecular aspects of the 5-day post-SE epileptogenic period. As in 5-day post-SE mice, we found that mTOR activity (as measured by pP70S6K$^{T389}$ levels), was significantly higher in HBP1$^{-/-}$ mice relative to HBP1$^{+/+}$ mice, (Fig 7B (FVBN background) and S6A Fig (C57BL/6 background)). Localization of pP70S6K$^{T389}$ in HBP1$^{-/-}$ mice by confocal imaging was also mostly neuronal, with high levels in HBP1$^{-/-}$ mice, while being nearly undetectable in HBP1$^{+/+}$ mice (Fig 7E). Metabolic re-arrangements predicted by the model of Fig 5F were also present in HBP1$^{-/-}$ mice. As seen in Fig 7F (left panel), the mRNA levels of the Wnt target PDK4, the regulator of PDH and TCA flux, was strongly elevated in both HBP1$^{+/-}$ and HBP1$^{-/-}$ mice. PDK4 phosphorylation of PDH at S293 was increased in the neurons of the HBP1$^{-/-}$

mice (Fig 7F, center and right panels). In results similar to the 5-day post-SE period (Fig 2D), elevated pPDH$^{S293}$ in HBP1$^{-/-}$ mice appeared to be localized to neurons in the hippocampus (Fig 7G). The effect of constitutive Wnt signaling in HBP1$^{-/-}$ mice also resulted in elevation of metabolic enzyme PKM2 and LDHA was also elevated, (Fig 7H). Finally, we observed HBP1$^{-/-}$ mice also had elevated PKM2 localized to astrocytes (Fig 7I). Together, these observations collectively demonstrate that increased Wnt signaling (through the deletion of one or both alleles of HBP1) drives the proposed metabolic and signaling pathways delineated for epileptogenesis. Specifically, increases in basal Wnt signaling resulted in regulation of PDK4 and PDH activity, altered Warburg metabolic enzymes and increased mTOR signaling. The results in Fig 7A–7I provide correlating genetic support for the proposed mechanisms from Figs 1–6 and underscore the role of Wnt signaling in driving metabolic and signaling changes.

The availability of a genetic mouse model with elevated Wnt signaling suggested a further test of the role of Wnt signaling in setting seizure thresholds, or how acquired seizures may arise. With the unexpected finding that the HBP1$^{-/-}$ mice exhibited spontaneous seizures, we hypothesized that the HBP1$^{+/-}$ or HBP1$^{-/-}$ mice may have elevated seizure susceptibility, or heightened seizure induction. If so, then lesser levels of kainate or pilocarpine may induce seizures in mice with deletion of one or both HBP1 alleles. Finally, considering the literature, there is a possible precedent, when viewed through the lens of Wnt signaling. One variation of the pilocarpine models uses lithium pre-treatment to efficiently induce SE with lesser doses of pilocarpine [22, 23]. Because lithium is an activator of Wnt signaling [22, 23], this well-used variation might be a clue that activated Wnt signaling facilitates seizure induction. Lastly, a major advantage is that the HBP1$^{-/-}$ and HBP1$^{+/-}$ mice were bred pure in C57BL/6 and FVBN backgrounds, so that seizure scoring and the doses with pilocarpine (FVBN) or with kainate (C57BL/6) can be reliably compared to wild-type controls.

As a test of the premise that Wnt signaling may regulate the seizure induction threshold, we treated HBP1$^{-/-}$, HBP1$^{+/-}$ and HBP1$^{+/+}$ mice in the FVBN background with pilocarpine (220 mg/kg see Methods) and found that no HBP1$^{-/-}$ mice survived seizure induction for more than an hour. Titration of the pilocarpine dose down to 150 mg/kg still demonstrated significant sensitivity for HBP1$^{-/-}$ mice (no survival past 1 day) while HBP1$^{+/-}$ mice had reduced survival compared to HBP1$^{+/+}$ mice (Fig 7J). These results suggested a clear gene-dosage effect on survival. Further titration of pilocarpine dosing to 135 mg/kg resulted in sufficient survival to observe a clear gene dosage effect for HBP1 genotypes, with HBP1$^{-/-}$ mice presenting with the highest number of severe seizures (Racine Scale 5) relative to HBP1$^{+/+}$ mice, with HBP1$^{+/-}$ mice having an intermediate in severity (Fig 7K). Similar results are seen for seizure induction by kainate (S6B Fig). Thus, the increased sensitivity to seizure induction in the HBP1$^{-/-}$ and HBP1$^{+/-}$ mice implicates Wnt signaling and a role for HBP1 as part of mechanisms for setting a seizure threshold and thus integral to epileptogenesis.

## Discussion

### Summary of results

The results of this paper provide new insights into the molecular and cellular basis of epileptogenesis—which is a necessary prelude to considering new therapeutic strategies to prevent chronic seizure onset. We utilized well-established models of temporal lobe epilepsy (TLE) (pilocarpine or kainate models), and other tools with approaches refined in other diseases to examine the period of early epileptogenesis. The period that immediately follows status epilepticus (SE) is surprisingly ill-characterized, though there are many beautiful metabolic studies for later periods surrounding chronic seizure onset in these models by Sonnewald and others [65, 68]. A comprehensive model supported by the data is presented in Fig 5F and therapeutic

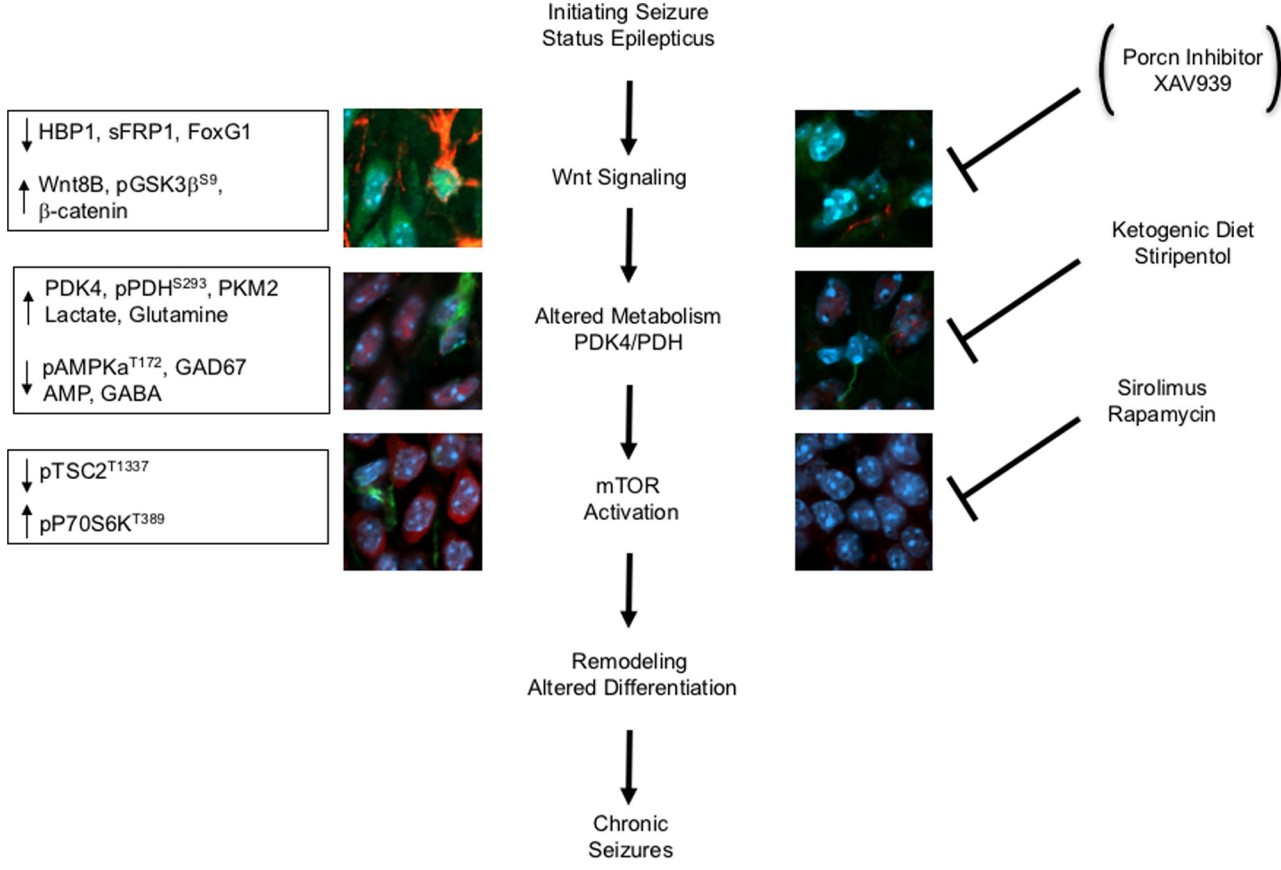

**Fig 8. A summary scheme for potential drugs to target the Wnt-AMPK-mTOR pathway during epileptogenesis.** Our epileptogenic mechanism model shows three target points for potential anti-epileptogenic drugs. First, a Wnt inhibitor may target the upstream of this pathway, but current treatments do not cross the BBB. Secondly, altered metabolism can be targeted by ketogenic diets and Stiripentol. Thirdly, mTOR activation can be targeted with mTOR inhibitors (e.g., Sirolimus and rapamycin). Collectively, the ideal therapy is a combination and nontoxic therapy aimed at multiple targets to alter remodeling and neuronal differentiation during the epileptogenic period to forestall the progression of chronic seizures.

hypotheses are summarized in Fig 8. But the present work goes further to illuminate a mechanism by which Wnt signaling triggers a unique and intricate metabolic reprogramming with downstream consequences known to be associated with epileptogenesis. The studies in the experimental models were complemented by a genetic model of constitutive Wnt signaling based upon deletion of the HBP1 gene, a known suppressor of Wnt signaling. Our first studies in the HBP1 mouse models show that these mice have spontaneous seizures and a heightened sensitivity to seizure induction by both pilocarpine and kainate. Furthermore, the HBP1 gene itself is expressed in the CA1 hippocampal region attributed to seizure control and is diminished in early epileptogenesis in the pilocarpine and kainate models at the peak of Wnt signaling. The experimental observations of Fig 7 suggest there is a remarkable concordance with the predictions of how HBP1 knockout mice with constitutive Wnt signaling and the pilocarpine/kainate models functionally overlap. Both the genetic and induced models exhibit the same features of metabolic rearrangement, including the induction of PDK4, the subsequent phosphorylation and inhibition of PDH and a robust mTOR activation. Together, these results suggest that HBP1 and Wnt signaling may be part of general mechanisms that alter seizure susceptibility during epileptogenesis.

## Wnt signaling and epileptogenesis

In general, active Wnt signaling is a necessary feature of stem cell renewal, proliferation and subsequent tissue development. Wnt signaling plays a role in numerous aspects of brain development (reviewed in [46]). Aberrant Wnt signaling has been linked to some genetic epilepsies by disrupting the orderly progression of normal developmental processes. A comprehensive model supported by the data figures is presented in Fig 5F. We documented robust Wnt signaling peaking at day 5 post status early in epileptogenesis, supported by several criteria: 1) GSK3β inhibition (Fig 1B and 1E); 2) β-catenin induction (Fig 1B and 1F); and 3) target gene activation (Axin2, PDK4; Figs 1C and 2A and 2B). The likely mechanism for an induction of Wnt signaling is a decrease of pathway inhibitors (HBP1, sFRP1, FoxG1) and induction of at least one Wnt ligand (Wnt8B). We selected HBP1, FoxG1 and Wnt8B because of previous relevance to neuronal development and/or regulating Wnt signaling. Other work has reported that Wnt signaling often occurs in neurons in different neurological contexts, but that Wnt ligands are often produced by astrocytes [78]. Thus, while the Wnt signaling phenotype is clear, we have not yet identified the (likely) astrocyte-derived Wnt ligands, of which there are at least 19 candidates. Our work suggests that Wnt8B is a plausible candidate (Fig 1C). Lastly, the role of Wnt signaling was additionally supported from a mouse model of constitutive Wnt signaling in which the pathway inhibitor HBP1 was deleted in the germline (HBP1$^{-/-}$ mice, Fig 7B–7D). Our lab discovered and characterized the HBP1 transcriptional repressor in the context of cellular senescence and Wnt signaling in breast cancer and other functions [29, 44, 79]. Numerous studies highlighted a role for HBP1 and a cognate microRNA in cellular senescence and in regulating Wnt signaling [41–44]. For neuronal contexts, a previous study reported that HBP1 expression regulated the progression of cortical development, similar to mechanisms in non-neuronal tissues. In fact, this work nicely developed our findings from a cell culture model into an elegant mouse model to highlight a role for the HBP1 gene in cortical development [40]. We are currently investigating the role of HBP1 in genetic epilepsies, as a case report highlighted the genetic loci at 7q22.3 in which HBP1 resides [80]. For all the above reasons, we explored the role of HBP1 in seizure generation via the mouse knockout of HBP1. The results of loss of one or both HBP1 alleles, as predicted, increased Wnt signaling (Fig 7B–7D, and S6A Fig). Furthermore, downstream targets such as PDK4 and mTOR were also activated in concordance with our model (Figs 5F and 7B and 7E–7G). Further, loss of one or both alleles of HBP1 resulted in a significant change in the sensitivity for seizure stimulation by either kainate or pilocarpine (Fig 7J and 7K, S6B Fig). Therefore, we hypothesize that the decline of HBP1 in early epileptogenesis and co-incident increased Wnt signaling may similarly alter the threshold for the development of subsequent chronic seizures, regardless if Wnt signaling was triggered by chemicals (e.g., pilocarpine or kainate) or deletion of HBP1.

In this study, we show metabolic re-programing that is triggered by Wnt signaling and the mechanisms leading to downstream activation of mTOR. We suggest that heightened Wnt signaling and metabolic reprogramming in early epileptogenesis (day 5 in the experimental model) may be important for setting up the aberrant developmental profile leading to recurrent seizures, of which heightened and transient stem cell and progenitor proliferation may be just one factor. First, numerous studies have documented a burst of progenitor cell proliferation following seizure-induced cell death, particularly in the sub-ventricular zone and hilus (reviewed in [81]). The resident stem-like cells in this region may sustain Wnt signaling as part of a stem-cell maintenance program that is present in control hippocampus, but Wnt signaling is greatly increased in the epileptogenic hippocampus. However, the proliferation of relatively few stem cells cannot account for the large metabolic response (exemplified by lactate production) coordinated by both neurons and astrocytes in a transient period around day 5. Rather,

we speculate that the metabolic reprogramming may be a contributing element of the aberrant neurogenesis and differentiation that has been advanced by Parent and others [52, 82]. As noted, while some new neurons are induced during this period in the sub-ventricular zone, the broad molecular changes we observe in almost all neurons are unlikely to be a result of growth demands. While a cancer cell uses Warburg metabolism to sustain the proliferation needs, the small of amount of aberrant proliferation in epileptogenesis cannot account for the large metabolic response, especially in lactate production (Fig 4A).

## Wnt signaling drives an unusual metabolic reprogramming—The central role of regulating pyruvate dehydrogenase

Using two different but complementary models, the major findings of this paper underscore that Wnt signaling in the brain orchestrates 1) regulation of a key metabolic gatekeeper enzyme pyruvate dehydrogenase (PDH), a known gene in genetic epilepsies [83, 84]; and 2) an unusual Warburg-like reprogramming that is accomplished in concert by astrocytes and neurons. While the mechanisms share some similarity with other diseases, there are mechanistic details that appear unique to brain and to the biochemistry and genetics of seizures. However, the remarkable similarities in the Wnt signaling triggered metabolic regulation, both in the status-induced and the HBP1$^{-/-}$ mouse models, suggest that we have identified a fundamental mechanism that may govern seizure susceptibility thresholds. Further, we believe that the metabolic aspects share similarities to a rare genetic metabolic disorder with accompanying epilepsy called Pyruvate Dehydrogenase Deficiency (PDHD) [84].

The results of this paper underscore the importance of regulating PDH activity through pyruvate dehydrogenase kinase (PDK). PDH and PDKs are major gatekeepers of carbon flow from the glycolytic product pyruvate into the Tricarboxylic acid (TCA) cycle. PDH and PDKs are thus determinants of aerobic glycolysis/Warburg metabolism. PDKs inhibit PDH by phosphorylation at serine 293, thus decreasing the flux of pyruvate into the TCA cycle and forcing glucose carbons into alternate fates such as conversion to alanine or lactate. Thus, the regulation of PDH is critical to determining the fate of the 3-carbon pyruvate for oxidative metabolism (acetyl CoA; TCA cycle) or for non-oxidative metabolism (lactate; Warburg effect).

First, our work in both genetic and induced mouse models with elevated hippocampal Wnt signaling (HBP1$^{-/-}$ mice and SE-induced mice) show that a key Wnt target gene is PDK4, which encodes an isoform of pyruvate dehydrogenase kinase. Waterman et al. first identified PDK1 as a LEF/TCF (lymphoid enhancer factor/T-cell factor; Wnt transcription factor)—regulated gene in cancer, but also noted a similarly significant regulation of PDK4 in a supplemental figure [24]. PDK1 regulation of PDH activity constituted an important nexus for Warburg re-programming in tumor cells by controlling carbon entry into the TCA cycle ([24, 25] and see below). Our work shows the PDK4 gene is induced in early epileptogenesis (day 5), while PDK1 was not activated (S2A Fig, Fig 2A and 2B). Further, the relevant PDK site on PDH (serine 293) is correspondingly modified in epileptogenesis (Fig 2C and 2D). Interestingly, in the HBP1 knockout model of constitutively active Wnt signaling, we also observed a significant increase in PDK4 in both HBP1$^{+/-}$ and HBP1$^{-/-}$ mouse hippocampus (Fig 7F), along with a similar but constitutive increase in pPDH$^{S293}$. Both the genetic and SE-induced experimental models with Wnt-driven PDK4-PDH phosphorylation appear to trigger downstream metabolic changes. PDK4 thus appears to be a uniquely and functionally important Wnt target gene in early epileptogenesis, coordinating a Warburg-like reprogramming in the hippocampus and through the regulation of PDH phosphorylation and inhibition.

We further asked whether changes other than the regulation of pyruvate dehydrogenase accompanied the robust lactate response (Fig 4A). Observations from tumor cells suggest that

high glucose flux and alterations in glycolytic enzymes often accompany a Warburg effect. In contrast to tumors, where the metabolism is confined within a single cell, the overall Warburg reprogramming in the hippocampus is accomplished in a concerted manner in both neurons and astrocytes. We have termed this unusual phenomenon as a "split" Warburg effect. Confocal microscopy demonstrated that the Warburg reprogramming in early epileptogenic hippocampus and in the HBP1[-/-] mice is uniquely partitioned between both neurons and astrocytes (Figs 2, 3C, 7G and 7I). These results are consistent with the split metabolism hypothesis, in which there is lactate flux from astrocytes to neurons. This hypothesis has been debated for years, but has begun to garner experimental support (reviewed in [25, 61, 85]). Our work firmly supports this novel configuration. The observed split Warburg metabolism is consistent with Wnt signaling that is also transiently and broadly induced in neurons and in a subset of astrocytes (Fig 1B–1F). PKM2, the Warburg isoform of pyruvate kinase, was induced exclusively in the bulk of astrocytes (Fig 3C), suggesting astrocytes engage in aerobic glycolysis. In other tissues, PKM2 and Wnt signaling are often co-activated ([86]). We have not investigated the role of PKM2 in epileptogenesis, but our metabolomics data in the pilocarpine model indicate a high glucose flux and robust production of lactate from the 3-carbon pyruvate. Notably, PKM2 is also abundant in HBP1[-/-] mice (Fig 7I). The LDHA isoform is also induced (Fig 3B), LDH has been reported to be a target in seizures [87] and a known agent for seizure treatment was reported to inhibit LDH, suggesting that lactate production may be important in seizure generation.

How is the altered biochemistry we observe during epileptogenesis related to the development of epilepsy? A key concept is the importance of PDH function, illustrated by the observation that PDH deficiency, in which one allele of the PDH alpha subunit is non-functional, results in a genetic epilepsy that manifests at birth with cerebral hypotrophy and defects in brain development [84]. PDH deficiency is commonly associated with lactate acidosis in brain and other areas in the affected patients, who often die in early childhood or infancy ([84, 88], reviewed in [89]).

Pascual and coworkers have reported a mouse model in which there is brain-specific reduction of PDH in heterozygous mice with a conditional knockout of PDHA1 (PDHA1[+/-], [88]). Importantly, this mouse model mimics aspects of human PDH-deficient epileptic encephalopathy, including dysfunctional excitability that was attributed to the reduced metabolic flux and lactic acidosis that resulted from the engineered PDH deficiency. These human and mouse studies implicate a metabolic etiology to certain genetic epilepsies and the importance of PDH regulation. In our studies, we identified alterations of PDH complex phosphorylation and activity through Wnt signaling-dependent expression of PDK4. Our studies thus hint that altered PDH activity may not be strictly relegated to rare genetic epilepsies, but could be a general feature of epileptogenesis. Together, the biochemistry and genetics underscore a unique role for PDH and the potential importance of Wnt signaling and its target gene PDK4. These mechanistic observations and the existing literature highlight a metabolic plasticity in early epileptogenesis and the critical role of PDH. Future investigations will be required to examine if there is a similarly central role for PDH in other epilepsies and seizure models.

## The consequences of altered metabolism in epileptogenesis: Alterations in neurotransmitter balance and mTOR activity

What might be the consequences of the Wnt signaling and metabolic mechanisms of Fig 5F? We consider two consequences: neurotransmitter balance and mTOR activation, which are two iconic consequences of epileptogenesis. A consequence of altered glucose metabolism can be a change in the glutamine-glutamate-GABA homeostasis (coupled through the TCA cycle).

In general, some metabolic disorders have been associated with abnormal and decreased corti-cal activity, while interspersed with periods of hyperexcitability, or seizures, possibly attributed to changes in neurotransmitter balance [84, 88, 90]. In our studies, we show that the early epi-leptogenic period, epitomized by day 5 post-SE, is characterized by an alteration in astrocyte metabolism with an increased glucose-to-lactate conversion and the appearance of Warburg metabolic enzymes in astrocytes. While the alteration of the TCA cycle (through PDH) will clearly influence the changes in glutamate-glutamine-GABA homeostasis of day 5 (Fig 4, [66]), additional enzymatic changes were also observed that contributed to altered homeostasis (Fig 4E and 4F). The metabolic and interconnected signaling defined in the early epileptogenic period are different than metabolic changes that occur at the onset of chronic seizures (14 days post-SE) and in the fully chronic period (4+ weeks post-SE). As Sonnewald et al. have elegantly described, hippocampal glucose flux through the TCA into glutamate, and derivatives gluta-mine and GABA, was decreased by 14 days. The decline in glucose flux remained into the chronic period ([68, 91, 92], reviewed in [65]). Together with the studies of Sonnewald and others, our results now provide a complete picture for early and late epileptogenesis. Our stud-ies highlight a possible and critical early therapeutic window in which signaling or metabolic interventions may obviate chronic seizure onset (see below).

mTOR activation is virtually iconic for seizure generation in human disease and numerous pre-clinical models [4, 8–10]. The surprising result in our studies is that the mTOR activation is firmly dependent on the Warburg-like metabolic re-arrangement that is triggered by Wnt signaling. We show clearly that the metabolic inhibitor 2-deoxyglucose blocked the 5-day post-status changes in AMPK and mTOR (Fig 6C), but did not affect Wnt signaling (Fig 6A) in the pilocarpine model. The glycolytic Warburg metabolism resulted in decreased AMP lev-els and decreased AMPK phosphorylation of TSC2 (Figs 4A and 5A). The net effect is the mTOR activation (Fig 5A) that has been previously observed in epileptogenesis [4]. Our stud-ies thus establish an interconnected Wnt-AMPK-mTOR pathway and their possible functions in epilepsy and suggests that altered metabolism has an important role in epileptogenesis [93, 94].

## Wnt signaling and metabolic reprogramming: Therapeutic implications for epilepsies

How might the metabolic and signaling alterations during epileptogenesis influence the devel-opment of epilepsy and effectuate new therapeutic strategies? An emerging theme of our stud-ies is that the hippocampus undergoes metabolic plasticity that may correlate and characterize epileptogenic transitions. Recent studies have revealed metabolic changes in neuronal and astrocytic differentiation [95]. Glycolytic states are often associated with undifferentiated pro-genitor cells in brain and other tissues. For example, the switch from PKM2 to PKM1 marks the transition from aerobic glycolysis in neural progenitor cells (NPC) to oxidative phosphory-lation in more differentiated neuronal cells [95]. Together with the established literature on mTOR and cellular remodeling, our mechanisms provide a plausible integration of Wnt sig-naling to mTOR activation through a metabolic reprogramming in early epileptogenesis and towards a brain that sustains chronic seizures.

Thus, our results suggest both a therapeutic window in early epileptogenesis before chronic seizures establish and predict several therapeutic intervention mechanisms. Our studies sug-gest that Wnt signaling in the epileptogenic period initiates a transient metabolic reprogram-ming that may play a role in the aberrant neurogenesis and differentiation to influence seizure sensitivity. This conclusion was reinforced by constitutive Wnt signaling through knockout of Wnt repressor HBP1, which results in constitutive induction of metabolic enzymes and

mTOR, and a heightened sensitivity to seizure induction by kainate and pilocarpine. In theory, blocking Wnt signaling should be very effective in preventing the signaling and metabolic cascade that was revealed by our studies. A recent study used a pre-clinical Wnt pathway inhibitor (XAV939) to block cellular remodeling associated with epileptogenesis induced by intrahippocampal injection of kainate [96]. An exhaustive analysis of Wnt target genes to verify a change in Wnt signaling was not done. In addition, the cellular analysis of this work then focused to 2-weeks following kainate induction, well after the changes detected in our work. We did not observe any effects with Wnt signaling inhibitors or tankyrase inhibitors (represented by XAV939) in our studies, but a major difference may be that the above 2019 study used intrahippocampal kainate injection, whereas our study used intraperitoneal injection of kainate. The brain-based injection of kainate in the published work may have provided a sufficient breach of the blood-brain barrier (BBB) to allow artificial entry of XAV939, however, Wnt pathway inhibition is likely to be difficult in humans as one inhibitor discussed below (XAV939) is not in clinical trials. Other Wnt pathway inhibitors (e.g., porcupine inhibitor LGK974) are in clinical trials for cancers, yet this compound is marginally or not bioavailable to the brain [97]. As the pharmacology of inhibiting Wnt signaling advances, more options may become available, though a cautious approach is required since some level of Wnt signaling is required for normal tissue homeostasis such as regeneration of gut epithelia.

The definition of altered glucose metabolism as an upstream signal into mTOR may provide better therapeutic opportunities centered on a metabolic etiology (see below). Our findings are most consistent with an activation of Wnt signaling that reprograms the hippocampus with a split Warburg metabolism that alters glucose metabolism in both neurons and astrocytes. These complex metabolic rearrangements then trigger a robust mTOR activation in response to the altered metabolism. Our studies have focused on transient changes at day 5 of epileptogenesis in the well-established pilocarpine or kainate models of TLE. This period has not been well-studied. The elegant studies by Sonnewald and coworkers focused on later periods of epileptogenesis, identifying an oxidative metabolism just prior to the onset of chronic seizures. Indeed, Rho and colleagues have suggested that considering epilepsy as a metabolic disease may provide a framework for discovering new treatments [98]. A clear hypothesis generated by our results suggests blocking the changes in epileptogenesis that we defined here might also block chronic seizure onset and chronic epilepsy. Thus, our studies predict that the early epileptogenic period after SE may represent a window for efficacious therapeutic treatment, perhaps with judiciously selected combinatorial interventions (Fig 8).

The data from this paper plausibly link mTOR activation observed in epileptogenesis with the later appearance of mTOR activity in epilepsies [4, 99]. Currently, mTOR inhibitors are used in various epileptic contexts and with reasonable justification. Evidence from animal models of adult TSC or PTEN loss demonstrate that constitutive activation of mTOR is sufficient to cause recurrent seizures, and that treatment with mTOR inhibitor rapamycin reduces seizures [20, 100]. Intriguingly, long-term treatment with rapamycin decreases the production of lactate in organotypic cultures, although the mechanism is not clear [101]. In the pilocarpine mouse model of acquired epilepsy, increased mTOR activity occurs constitutively with the appearance of chronic seizures. mTOR inhibitors have been in clinical testing for treatment of seizures [102], but with varying levels of effectiveness. Treatment with rapamycin is also sufficient to suppress chronic seizures. However, removal of rapamycin resulted in reacquisition of seizures, suggesting that there is an underlying cause for both mTOR activity and seizures [103]. With the mechanistic framework of this paper, there are new opportunities to discover collaborative therapeutic strategies.

The metabolic plasticity in early epileptogenesis suggests that diminishing glucose usage might result in diminished mTOR activity, perhaps increasing the efficacy of existing

inhibitors to lesser doses and which, in turn, limit undesirable side-effects. Numerous metabolic interventions have been discussed and reviewed [87, 104, 105]. A key observation in Fig 5 is that treatment with 2-deoxyglucose (2DG), suppresses the mTOR activation, but not the Wnt signaling—thus firmly sequencing the metabolic changes between Wnt signaling and mTOR activation. Numerous and convincing studies by Stafstrom and coworkers in pre-clinical models have suggested intervention with 2DG [74, 75]. However, 2DG was discontinued in cancer clinical trials when cardiac toxicity (i.e., grade 3 asymptomatic QT prolongation) was observed in prostate cancer patients following prolonged treatment [106]. The discontinuance in clinical trials for advanced cancer should give pause in the use of 2DG in humans.

We suggest two possibilities: ketogenic and/or low glycemic diets and an approved drug Stiripentol. As an alternative metabolic intervention to the more toxic 2DG, ketogenic diet (KD) and low glycemic diets have been widely discussed and used of treatment of refractory epilepsies [107], although less so in adults [108]. The ketogenic diet is a high fat, highly restricted carbohydrate and protein diet that results in a significant decrease in blood glucose and production of ketone bodies, which are utilized by neurons in place of glucose [109]. While the KD is characterized by the presence of ketone bodies, the low-glycemic diet does not exhibit ketosis. There has also been debate as to whether ketone bodies affect seizure generation (reviewed in [105]). Nevertheless, dietary/metabolic interventions have been effective, but compliance can be difficult, since lapses in the KD can result in reacquisition of seizures [110]. Based on the present work, an immediate post-status temporary course of the KD, or perhaps a low glycemic diet might be predicted to attenuate the early metabolic changes and subsequent sequalae. Indeed, post-status ketogenic diet results in a modest reduction of hippocampal mTOR signaling following kainate-induced SE (measured at 3- and 7-days post-SE) [111]. Additionally, ketone bodies such as 3-hydroxybutyrate inhibit AMPK activity, similar to what we observe in Fig 6C [112–114]. Thus, the altered metabolism, decreased AMP and strong AMPK inhibition in the epileptogenic period suggest that the efficacy of the ketogenic diet could function by opposing the metabolic changes documented here. Nonetheless, the KD or low glycemic diets achieve the goal of diminishing glucose uptake and usage and may set a therapeutic stage to lessen use of mTOR inhibitors to treat epilepsies.

Another intriguing possibility might be the use of Stiripentol or derivatives to block lactate production, inhibiting the consequences of our metabolic mechanism. While initially aimed at increasing GABAergic activity, the published studies highlighted that the seizure attenuation by Stiripentol was likely due to a blockade of lactate dehydrogenase to complement the GABAergic effects [87, 98]. Stiripentol (Diacomit) is FDA approved as adjunctive therapy in Europe and the USA for the treatment of Dravet's syndrome, a genetic epilepsy (reviewed in [115]). Intriguingly, the KD also has efficacy in some intractable Dravet's patients, perhaps suggesting a metabolic etiology to this epilepsy. Notably, our mechanisms feature a robust burst of lactate production (Fig 4A and 4D), so this may be a mechanism-based pharmacological intervention. Thus, a short-term course of Stiripentol following the initiating seizure may have more immediate effect than KD in obviating the induced Warburg metabolism, downstream mTOR activation, and the cellular and molecular changes that may contribute to chronic seizure onset.

With multiple points for intervention as outlined in Fig 8, we propose taking therapeutic strategy cues from the cancer field and develop combinatorial strategies based on the results of biochemical and cellular mechanistic elaboration of the early epileptogenic period to prevention of chronic seizures. As noted, early intervention with KD has efficacy in reducing seizures and mTOR activation. Thus, immediate post-SE metabolic intervention combined with mTOR inhibitors in the early epileptogenic period may have unique and greater efficacy in preventing the onset of chronic seizures.

In conclusion, the work of this paper elaborates new molecular mechanisms and frame-works for early epileptogenesis, initiated by Wnt signaling, resulting in a novel Warburg metabolic reprogramming and then signaling into mTOR. The Wnt-driven signaling and metabolism program during status epilepticus may represent a therapeutic window to prevent the onset of chronic seizures. Remarkably, the unbiased metabolomic and molecular analyses are fully consistent with the transient appearance of a Warburg metabolism (though split between different cell types). Support for the role of Wnt signaling in driving the observed changes derives from a mouse model with constitutive Wnt signaling in the hippocampus, driven by knockout of the Wnt repressor HBP1. Further, our work links metabolic changes to the alterations in the glutamine-glutamate-GABA homeostasis that characterizes seizure susceptibility. While we have discussed our results in the context of the cell biology and of possible therapeutic implications, our work additionally advances a robust molecular platform and roadmap for discovering new therapeutic strategies or combinations, perhaps by pharmaco-logical and/or metabolic interventions in the early period of epileptogenesis. Clearly, future work will be required to determine if therapeutic intervention during the early epileptogenic window defined in this paper attenuates the onset of chronic seizures in epilepsies.

## Materials and methods

### Mouse strains

The two mouse strains used in these experiments were pure background FVBN (The Jackson laboratory) and C57EL6 (The Jackson laboratory) strains. Each strain was bred to include an HBP1 knockout allele caused by the insertion of the B-galactosidase gene into exon 2 of HBP1. Mice could be bred to be wild type (WT), heterozygous (HET), or knockout (KO). By mating HET mice, a Mendelian spread of all three genotypes was obtained. To limit individual variation, only male mice were used for these experiments.

### Seizure induction and drug administration

7-week old male C57BL/6 (The Jackson laboratory) mice were injected intraperitoneally with 30 mg/kg kainate (Milestone PharmTech) in neutral saline. Similarly, 7-week old male FVBN (The Jackson laboratory) mice were injected intraperitoneally with 220 mg/kg pilocarpine (Sigma-Aldrich) in neutral saline. Within 30 minutes, the injected mice developed seizures that were sustained for a minimum of 1 hour. The resulting seizures were scored using the Racine scale [73]. Mice were selected for further analysis if they showed a minimum of two Racine 5 seizures. Mice were supported as needed by intraperitoneal injection of neutral saline and with soft food for a minimum of 48 hours. All animal experiments were carried out in accordance with guidelines approved by the Institutional Animal Care and Use Committee (IACUC) at Tufts University (Boston, MA). Pilocarpine- or kainate-treated mice were euthanized between 3 and 7 days. The brain regions were dissected on ice, then frozen in nitrogen for later protein, RNA or metabolite analysis. Control and pilocarpine mice were treated with 2-deoxyglucose 250 mg/kg [74] once daily from day 1 to day 5 post-SE. For glucose flux metabolite analysis, mice were injected intraperitoneally with 500 mg/kg [U-$^{13}$C] d-glucose (Cambridge Isotope Laboratories) in neutral saline 15 minutes prior to euthanasia. For seizure sensitivity experiments in HBP1 mice, SE was induced through intraperitoneal injection of pilocarpine in the FVBN strain and kainate in the C57 strain. Pilocarpine initially was injected at 150 mg/kg, then at 135 mg/kg to minimize the death rate. Kainate was injected at 30 mg/kg. Control mice were injected with neutral saline in order to obtain baseline levels of brain biochemistry.

## Quantitative RT-PCR

The primers used here are detailed in the Supplemental Material. Total hippocampal RNA was extracted using RNAeasy Mini kits (Qiagen) and the reverse transcribed with random hexamer mix (iScript, Bio-Rad). The resulting cDNA was used for real-time qPCR performed using an iCycler (BioRad). SYBR-Green master mix was purchased from Bio-Rad. All quantifications were normalized to an endogenous 18 S RNA control. The relative quantitative value for each target gene compared to the calibrator for the target is expressed as comparative Ct ($2^{-(\Delta Ct-Cc)}$) method (Ct and Cc are the mean threshold cycle differences after normalizing to 18S). Quantitative RT-PCR experiments were performed in triplicate and the percent coefficient of variation for the set value was less than 0.1. PCR efficiency with given primers was between 95 to 105%.

## Western blotting

Whole tissue lysates were prepared from frozen tissue by extraction for 30 min at 4°C in lysis buffer (50mM HEPES, pH7.5, 150 mM NaCl, 1.5mM $MgCl_2$, 1MM EDTA, 10% glycine, 1% Triton X-100, 20mM β-glycerophosphate, 0.1 mM sodium vanadate, 0.1% SDS, 1% deoxycholate, 1 mg/ml leupeptin, 200 mM PMSF, 1 mg/ml pepstatin). All lysates were clarified by centrifugation for 15 min at 14000 rpm and frozen at –80°C. The supernatants were quantified by using Quant-It Assay protocol from Thermo Scientific. 30 µg of total protein samples in each group were subjected to SDS-PAGE (Bio-Rad Mini-PROTEAN TGX gel), then transferred to PVDF membrane. The membranes were incubated with primary antibodies overnight at 4°C and secondary antibody for 1 hour at room temperature. The antibodies used were provided in Supplemental Material. The western blot results were developed with Thermo Fisher Supersignal West Femto Chemiluminescent substrate and quantitated by the Bio-Rad Chemidoc MP Imaging System.

## NMR analysis of metabolites

Dissected and frozen hippocampus was finely minced on dry ice and extracted in 0.4 mL of ice-cold methanol and 0.085 mL water. After sonication to homogeneity, 0.4 mL of cold chloroform was added and further processed as described [116]. The polar methanol/water layer was dried in a speed-vac and then reconstituted in 50 mM phosphate $D_2O$ buffer pH 7.0 with 0.2 mM of NMR standard (DSS, 4,4-dimethyl-4-silapentane-1-sulfonic acid). $^1$H NMR spectra of each sample were collected at 25°C on a Bruker Avance 600 spectrometer using 256 scans and a NOE1D pulse sequence. Data on $^{13}$C-labeled samples used a ZGDC pulse sequence and 12,000 to 20,000 scans. The $^1$H data were processed and analyzed using CHENOMX NMR Suite 8.0 to identify and quantify compounds present and to measure their concentrations. Compound identities were also verified by 2D $^{13}$C-$^1$H HSQC spectroscopy. Peak concentrations of metabolites were input into Metaboanalyst, normalized to the sum (scaled using the Pareto method), and differences between groups were analyzed for statistical significance [117]. A dendritic heatmap was generated using the Pearson distance measure and Average clustering method in Metaboanalyst 4.0. Enrichment using the curated pathway-associated metabolite sets was performed in Metaboanalyst 4.0. In Fig 4D, the individual metabolite carbons labeled by [U $^{13}$C]-glucose, identified in S4D Fig and quantified in S2 Table, were sum normalized and scaled using the Pareto method. Then, a dendritic heatmap was generated using the Pearson distance measure and Average clustering method in Metaboanalyst 4.0.

## Immunofluorescence staining

7-week old male C57BL/6 or FVBN wild type mice were sacrificed after 5-days following pilocarpine injection and induced status epilepticus (see above). The mice were anesthetized with

isoflurane 4% and perfused with cold 4% PFA (pH 7.4). The brains were surgically removed and fixed in cold 4% PFA for at least four hours or overnight at 4°C. The fixed brains were cryoprotected in 30% Sucrose overnight at 4°C. Hippocampal coronal sections (12 μm) were cut using a Leica Biosystem cryostat. The coronal sections were stored at -20°C for later immunostaining. Before immunostaining, the sections were pretreated with 3% hydrogen peroxide in methanol for 5 min and washed with PBS for another 5 min. Thereafter, the coronal sections were incubated with primary antibodies 1:100, as listed in Supplemental Material, diluted in 10% donkey serum (Abcam), 5% Non-fat dry milk, 4% BSA (Thermo Fisher), and 0.1% Trion X-100 (Sigma Aldrich) overnight at 4°C in moisturized chamber. The next day, the sections were washed with PBS for 5 minutes three times and incubated with secondary antibody 1:150; as listed in Supplemental Material, diluted in donkey blocking buffer at room temperature for one hour. Afterwards, the sections washed three times with PBS for 5 minutes and then mounted with flourished mounted DAPI (Abcam) for confocal imaging.

## Confocal imaging

Stained sections were imaged on a LSM800 laser scanning confocal microscope (ZEN 2.3-blue edition) in multi-tracking mode. The images were obtained with optimal pinhole a 63X, 1.4 NA, plan-apochromat oil immersion objective, 0.8 micron per optical sections, at a scan speed of 3, averaging of 4 images with an overall pixel dwell time of ~12 μsec. All image processing and analysis was performed in Fiji. In all images, only balance, contrast, and evenness of the illumination were adjusted. Final immunolabeling quantification and co-localization results were reported as percentage of positive cells (neuronal or astrocytes)/ total cells (neuronal or astrocytes). The quantification was conducted by three blinded investigators in four-six hippocampal coronal sections (section width = 12 μm) per mouse. Then, a minimum of 4 photographic fields were quantified per section (n = 3–4 mice for each group) using Fiji.

## Statistical analysis

Statistical analysis was performed using Student's unpaired $t$-test. $P < 0.05$ was considered statistically significant. All western blot, qRT-PCR and cell quantification data were analyzed using GraphPad Prism version 9.0 software. Metabolite Set Enrichment Analysis (MSEA), Principal Component and PLS analyses were done using Metaboanalyst 4.0 (www.metaboanalyst.ca).

## Supporting information

**S1 Fig. Wnt signaling during epileptogenesis in vivo studies.** (A) Wnt signaling is activated in the early epileptogenic period using kainate induction in mice. Status epilepticus was induced in male 7-week C57BL6 mice with kainate (30 mg/kg). Representative western blots and quantified data of hippocampal extracts from 3-, 5- or 7-day post-kainate (KA) or control saline (Sal) are shown for pGSK3β$^{S9}$, GSK3β, β-catenin, and actin (loading control). (n = 4 mice for each group). (B) Wnt signaling is activated in the early epileptogenic period (pilocarpine induction) in rats. Status epilepticus was induced in 4–5 male 7-week Wistar rats (Charles River) with pilocarpine (300 mg/kg). Representative western blots of hippocampal extracts from different time 3-hours, 24-hours, 3-days, or 7-day pilocarpine (Pilo) or control saline (Ctr) are shown for pGSK3β$^{S9}$, total-GSK3β and tubulin (loading control)-right. Quantified data from the western blots from pilocarpine-treated and saline-treated is shown (left). (C) Expression of Wnt pathway genes. qRT-PCR of Wnt regulators expression in hippocampus following kainate at day 5 post-SE compared to control mice (n = 4–6 mice for each group). Data are represented as mean ± SEM. $^{*}P <0.05$, $^{**}P < 0.01$ by unpaired student's $t$-test using

Prism 9.0 (Graphpad). (D, E) Localization of pGSK3β and of β-catenin in the hippocampus post-SE. pGSK3β[S9] (green; D) and β-catenin (green; E) in hippocampus at day 5 post-SE with pilocarpine compared to control mice [Dapi (blue), and NeuN (red)]. Percentage of neuronal and astrocytic positive β-catenin are increased at the DG region of 5-days post-SE with pilocarpine hippocampus when compared to control mice (see supplemental methods). (F) Localization of Wnt Signaling in the hippocampal region following SE induction. We examined the *in vivo* localization of SE-induced Wnt signaling in mouse hippocampus using the Wnt/β-catenin BAT-GAL reporter mouse [118]. Coronal sections from 5-day control and kainate-SE mice were stained for β-galactosidase activity to determine the localization of Wnt-dependent gene expression (Fig 1). In control mice, there is constitutive activity in the dentate gyrus (DG) but little signal elsewhere. Localization of signal is prominent in the granule cortical layer (GCL) with some activity in the subgranular zone (SGZ). These results are nearly identical to previous observations [37]. Five days after SE, Wnt signaling was strongly induced throughout the DG, including the SGZ and GCL. Wnt pathway activation is also present in the CA1 region. Refer to supplemental materials for more details on the number of replicates and statistical analyses for each experiment.
(PDF)

**S2 Fig. Expression of pyruvate dehydrogenase kinase 1 (PDK1) is not altered at day 5 post-SE.** Quantitative RT-PCR of PDK1 expression levels in hippocampus at day 5 following SE with pilocarpine (Pilo) or control (saline). (n = 5–6 mice/group). Data are represented as mean ± SEM. **$P$ <0.01, by unpaired student's *t*-test using Prism 9.0 (Graphpad). Refer to supplemental materials for more details on the number of replicates and statistical analyses for each experiment.
(PDF)

**S3 Fig. Status-induced Warburg metabolic enzymes.** (A) The mRNA expression of Warburg metabolic enzyme isoforms is elevated after kainate-induced SE. Quantitative RT-PCR detection of HK II, PKM2 and LDHA gene expression levels from the hippocampi of the 3-, 5- and 7-day post-status epilepticus mice in (kainate treated) compared to control (saline) mice. (B) The protein expression of Warburg metabolic enzyme isoforms is elevated after kainate-induced SE. Representative and quantified western blots of hippocampal extracts from 3-, 5- or 7-day kainate (KA), or control saline (Sal) are shown for PKM2, LDHA, and actin (loading control). (n = 3–5 for each group). Data are represented as mean ± SEM. *$P$ <0.05, **$P$ < 0.01 by unpaired student's *t*-test using Prism 9.0 (Graphpad). Refer to supplemental materials for more details on the number of replicates and statistical analyses for each experiment.
(PDF)

**S4 Fig. Metabolite analysis reveals elevated lactate and glucose metabolism in 5-day post-SE hippocampal metabolism—Signatures of warburg metabolism.** (A) Representative [1]H NMR spectra of hippocampal metabolites. Red: Control, Black: Pilocarpine, Green: Difference spectrum (scaled 4x). The NMR peaks corresponding to significantly changed levels of metabolites are labeled. A non-significant resonance at 3.716 ppm was removed for clarity. Non-standard abbreviations used are Suc, succinate; GPC, sn-glycero-3-phosphocholine; Myo, myo-inositol; OPC, O-phosphocholine; GABA, 4-aminobutyrate; NAA, N-acetylaspartate. (B) Partial least squares analysis demonstrates significant difference between control and 5-day post-SE metabolism. The metabolites showed significant separation (>95% confidence) between control mice (red) and pilocarpine-treated mice (green). (C) Metabolite changes correlated and anti-correlated with lactate in control and 5-day post-SE metabolism. The top 25 metabolites pro- and anti-correlated with lactate are shown along with their correlation

coefficients calculated in Metaboanalyst 4.0 [62, 63]. (D) Representative $^1$H-$^{13}$C HSQC NMR spectrum used for identification of hippocampal metabolites after infusion of [U-$^{13}$C]-glucose and administration of pilocarpine. The NMR peaks corresponding to significantly changed levels of metabolites are circled and labeled. Abbreviations used: Suc, succinate; GPC, sn-gly-cero-3-phosphocholine; Myo, myo-inositol; GABA, 4-aminobutyrate; NAA, N-acetylaspartate. (E) Quantitative RT-PCR of Pyruvate Carboxylase (PC) expression levels in hippocampus at day 5 following SE with pilocarpine (Pilo) or control (saline). (n = 4–6 mice/group). Data are represented as mean ± SEM. $P > 0.05$, by unpaired student's *t*-test using Prism 9.0 (Graphpad). Refer to supplemental materials for more details on the number of replicates and statistical analyses for each experiment.
(PDF)

**S5 Fig. Glycolytic inhibitor 2-DG does not alter PDH activity.** (A) Confocal images (X63) of positive pPDH$^{S293}$ (red) neuronal cells that are not changed in the DG region of 5 days post-SE in hippocampus 2-DG-treated pilocarpine mice when compared to control pilocarpine mice. [GFAP (green) and Dapi (blue)].
(PDF)

**S6 Fig. Induced seizure sensitivity in HBP1$^{-/-}$ KO mice for C57BL/6 strain.** (A) Representative western blot with quantified data of baseline levels of Wnt and mTOR signaling markers in C57BL/6 mouse strain were increased in HBP1$^{-/-}$ Knockout (KO) and HBP1$^{+/-}$ heterozygous (HET) when compared to control (WT). (B) HBP1$^{-/-}$ KO mice have increased number and severity of seizures after seizure induction based on the Racine scale [73] (n = 3–5 mice for each group). Data are represented as mean ± SEM. $^*P < 0.05$ by unpaired student's *t*-test using Prism 9.0 (Graphpad). Refer to supplemental materials for more details on the number of replicates and statistical analyses for this experiment.
(PDF)

**S1 Table. Hippocampal polar metabolites from control and 5-day pilocarpine-treated mice were extracted (4–5 hippocampi) and quantified by [$^1$H] NMR.** Metabolites that change significantly ($P < 0.1$) are shown.
(XLSX)

**S2 Table. Hippocampal polar metabolites from control and 5-day pilocarpine-treated mice injected with [U $^{13}$C]-glucose, were extracted (4–5 hippocampi) and quantified by [$^{13}$C] NMR.** Metabolites that change significantly ($P < 0.1$) are shown.
(XLSX)

**S1 File.**
(PDF)

**S2 File.**
(DOCX)

**S3 File.**
(PZF)

## Acknowledgments

*We thank the agencies and institutions (listed in the Financial Disclosure sections) that made this work possible. RA especially acknowledges her home university of* Umm Al-Qura University, Makkah, Saudi Arabia. We additionally thank Dr. Victor Hatini and his Tufts Confocal core for generous use, training and advice throughout this project. The images in this paper are a

testament to his generosity of time and access to confocal microscopy core. Finally, ASY and ASY especially acknowledge and thank Susan and David Axelrod for their unwavering support of our work; for taking a chance on this unusual multi-disciplinary partnership of a cancer biologist and a neurologist and most of all, for their instrumental leadership for the CURE foundation.

## Author Contributions

**Conceptualization:** Audrey S. Yee, Taylor Malone, James D. Baleja, Amy S. Yee.

**Data curation:** Roaya S. Alqurashi, Sumaiah Alrubiaan, Mary W. Tam, Kai Wang, Rozena R. Nandedwalla, Wesley Field, Dalal Alkhelb, Katherine S. Given, James D. Baleja, K. Eric Paulson, Amy S. Yee.

**Formal analysis:** Roaya S. Alqurashi, Audrey S. Yee, Taylor Malone, Sumaiah Alrubiaan, Mary W. Tam, Kai Wang, Rozena R. Nandedwalla, Wesley Field, James D. Baleja, K. Eric Paulson, Amy S. Yee.

**Funding acquisition:** Audrey S. Yee, Amy S. Yee.

**Investigation:** Roaya S. Alqurashi, Audrey S. Yee, Taylor Malone, Sumaiah Alrubiaan, Mary W. Tam, Kai Wang, Rozena R. Nandedwalla, Wesley Field, Dalal Alkhelb, Katherine S. Given, Raghib Siddiqui, James D. Baleja, K. Eric Paulson, Amy S. Yee.

**Methodology:** Roaya S. Alqurashi, Mary W. Tam, Kai Wang, Rozena R. Nandedwalla, Wesley Field, Raghib Siddiqui, James D. Baleja, K. Eric Paulson, Amy S. Yee.

**Project administration:** James D. Baleja, K. Eric Paulson, Amy S. Yee.

**Resources:** Amy S. Yee.

**Supervision:** K. Eric Paulson, Amy S. Yee.

**Validation:** Roaya S. Alqurashi, James D. Baleja, K. Eric Paulson, Amy S. Yee.

**Visualization:** Audrey S. Yee, K. Eric Paulson, Amy S. Yee.

**Writing – original draft:** Roaya S. Alqurashi, K. Eric Paulson, Amy S. Yee.

**Writing – review & editing:** Roaya S. Alqurashi, Audrey S. Yee, James D. Baleja, K. Eric Paulson, Amy S. Yee.

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
