## [Decision Letter · Decision Letter 0]

8 Sep 2020

PONE-D-20-22075

A Warburg-like Metabolic Program Coordinates Wnt, AMPK and mTOR Signaling Pathways in Epileptogenesis

PLOS ONE

Dear Dr. Yee,

Thank you for submitting your manuscript to PLOS ONE. After careful consideration, we feel that it has merit but does not fully meet PLOS ONE’s publication criteria as it currently stands. Therefore, we invite you to submit a revised version of the manuscript that addresses the points raised during the review process.

We look forward to receiving your revised manuscript.

Kind regards,

Giuseppe Biagini, MD

Academic Editor

PLOS ONE

Journal Requirements:

"This work was supported by grants from the Department of Defense (W81XWH-10-1-0381, ASY, ASY), the CURE Foundation (ASY and ASY) and by Umm Al-Qura University, Makkah, Saudi Arabia (RA). The work utilized NMR instrumentation that was purchased with funding from a National Institutes of Health SIG grant (S10OD020073)."

"ASY and ASY

Funder - Department of Defense

(W81XWH-10-1-0381)

ASY and ASY

Funder - Cure Foundation

www.cureepilepsy.org

(No Grant Number)

RA

Umm Al-Qura University, Makkah, Saudi Arabia

(Student Fellowship - no number)

The funders had no role in study design, data collection and analysis, decision to

publish, or preparation of the manuscript."

4. Please amend the manuscript submission data (via Edit Submission) to include author Katherine Given.

Reviewers' comments:

Reviewer's Responses to Questions

**Comments to the Author**

1. Is the manuscript technically sound, and do the data support the conclusions?

Reviewer #1: Yes

Reviewer #2: Partly

2. Has the statistical analysis been performed appropriately and rigorously? 

Reviewer #1: I Don't Know

Reviewer #2: Yes

3. Have the authors made all data underlying the findings in their manuscript fully available?

Reviewer #1: No

Reviewer #2: Yes

4. Is the manuscript presented in an intelligible fashion and written in standard English?

Reviewer #1: Yes

Reviewer #2: Yes

5. Review Comments to the Author

Reviewer #1: In their manuscript, Yee et al report that after status epilepticus (SE), an orchestrated metabolic change occurs which is governed by WNT-signalling and downstream mTOR activation, resulting in glycolisis shunting the Krebs cycle.

This paper is a very comprehensive report - commendably in many experiments based on two different SE models. Some issues, however, nedd to be clarified:

Major concenrs:

1. Overall, there is a strong mix of methods and results, with methods often being elaborately explained in the resultssection or the figure legends (which were integrated in the text and often split, which made reading difficult). The authors should try to dientangle this mixture as far as possible to increase legibility.

2. Similarly, also results and introduction are mixed to some degree: Lines 110-112 and again 123-126 are somewhat summing up the results within the Introduction section. Again, this should be avoided.

3. The discussion is too long and should be condensed.

4. Regarding the involvment of Wnt and SE, the findings of Gorter et al (214, in Neurobiology of Disease No. 62) should be acknowledged.

Minor issues:

Line 45: the sentence is a bit cryptic. Please rephrase.

Line 448: Data is plural, hence please use plural also in the verb.

Reviewer #2: The authors demonstrate a transient activation of the wnt-betacatenin pathway during the latent period following SE similar as found in some cancers. This line of research requires investigation but the results as presented are associative and the authors do not show causality. My comments are as follow:

a. I would suggest toning down statements such as saying this is the first to show the complex signaling and metabolic changes underlying epileptogenesis as the authors do not show causality. Also since a recent study by Wolff et al 2020 argues that the latent period may not even exist as subclinical seizures may continue during this period. This study potentially helps increase our understanding but without data showing that inhibition of wnt-betacatenin pathways leads to less spontaneous recurrent seizures following SE- these results remain an association. Especially since the changes are see only for one day. Further experiments are required to address this significant weakness of this study.

b. It is argued that wnt signaling inhibitors do not cross the BBB and therefore their effect was not tested or when tested was ineffective. Why not deliver these drugs icv? Especially since 2DG was ineffective in normalizing the SE Induced changes. I think this is a critical missing part of this study.

c. To increase the impact of this study it would be necessary to treat the animals with wnt-betacatenin inhibitors and demonstrate an effect on spontaneous recurrent seizures.

d. I don’t clearly see the link between the mTOR and GABA story. Please remove the GABA and glutamate results which can be part of another paper.

e. Methods (line 944): Please explain further what is meant by “ a minimum of 4 photographic field…per mouse). For example how many sections does this equate to. Was there attempts at a co-localization analysis? Was the person doing the counts blinded? Also please state how thick were your sections.

f. Methods Confocal (lines 942 and 945): Fiji and Image J are the same programs except the former has many already installed plugins.

g. Abstract: The following is repetitive and can be shortened “The results of this paper build a new molecular framework of complex signaling pathways for early epileptogenesis. Our framework advances the understanding of the complex molecular changes in early epileptogenesis and creates a detailed platform for discovering new future therapeutic strategies based upon Wnt signaling and/or metabolic interventions. Our studies may define a unique window in early epileptogenesis for attenuating recurrent and chronic seizures that define epilepsies.”

h. Introduction: Rapamycin also reduced seizures in a model of infantile spasms. Please add reference.

i. Introduction: May be useful to add figure which summarizes the relationship between wnt signaling and mTOR .

j. Introduction: Frizzled receptors are mentioned without context.

k. There are many abbreviations that require definition. Please provide a definition at their first appearance.

l. Introduction (line 114). Warburg metabolism is mentioned without a definition.

m. Introduction: (lines 115-116) …should say “that occurs in neurons” and not “…are occurs…”

6. PLOS authors have the option to publish the peer review history of their article (what does this mean?). If published, this will include your full peer review and any attached files.

Reviewer #1: No

Reviewer #2: No

---

## [Author Response · Author response to Decision Letter 0]

18 Mar 2021

Author’s response to Journal Comments

Journal. Please include the following items when submitting your revised manuscript:

Authors’ Response. Separate files of a cover letter, point-by-point rebuttal, a marked up copy, and original copy have been uploaded.

Journal. If applicable, we recommend that you deposit your laboratory protocols in protocols.io to enhance the reproducibility of your results. Protocols.io assigns your protocol its own identifier (DOI) so that it can be cited independently in the future. For instructions see: http://journals.plos.org/plosone/s/submission-guidelines#loc-laboratory-protocols

Authors’ Response. There are no changes in the financial disclosures.

Journal. When submitting your revision, we need you to address these additional requirements. 

 Authors’ Response. We have used this a guide for revision.

Authors’ Response. We had included all the images of uncropped blots in the supplemental data, but have labelled them more clearly. We also consider this to be an important part of substantiating the figures.

"This work was supported by grants from the Department of Defense (W81XWH-10-1-0381, ASY, ASY), the CURE Foundation (ASY and ASY) and by Umm Al-Qura University, Makkah, Saudi Arabia (RA). The work utilized NMR instrumentation that was purchased with funding from a National Institutes of Health SIG grant (S10OD020073)."

"ASY and ASY

Funder - Department of Defense

(W81XWH-10-1-0381)

ASY and ASY

Funder - Cure Foundation

www.cureepilepsy.org

(No Grant Number)

RA

Umm Al-Qura University, Makkah, Saudi Arabia

(Student Fellowship - no number)

The funders had no role in study design, data collection and analysis, decision to

publish, or preparation of the manuscript."

Authors’ Response. We have removed funding information from the acknowledgements.

This is now the correct funding information. 

ASY and ASY

Funder - Department of Defense

(W81XWH-10-1-0381)

ASY and ASY

Funder - Cure Foundation

www.cureepilepsy.org

(No Grant Number)

RA

Umm Al-Qura University, Makkah, Saudi Arabia

(Student Fellowship - no number)

NMR instrumentation (National Institutes of Health SIG grant (S10OD020073)."

The funders had no role in study design, data collection and analysis, decision to

publish, or preparation of the manuscript."

4. Please amend the manuscript submission data (via Edit Submission) to include author Katherine Given.

 Authors’ response. We have added Ms. Givens. 

 Authors’ response. Hopefully, the supporting information is now correctly configured.

 

Author’s Response to Reviewers

Reviewer’s Comments to the Author:

1. Is the manuscript technically sound, and do the data support the conclusions?

Reviewer #1: Yes

Reviewer #2: Partly

2. Has the statistical analysis been performed appropriately and rigorously? 

Reviewer #1: I Don't Know

Reviewer #2: Yes

3. Have the authors made all data underlying the findings in their manuscript fully available?

Reviewer #1: No

Reviewer #2: Yes

Authors’ Response to points 1, 2, and 3: We have edited the manuscript to include precise descriptions of the sample sizes and underlying data to determine the statistical significance of all our findings. Throughout our studies, we emphasized statistical rigor in determining conclusions based upon our experimental data. We regret that that this very important point was not completely clear to both reviewers and have now tried to improve the clarity. All statistical analyses were performed in the latest version of Prism. The relevant calculation files are now provided in Supplemental Information. In the unrevised version, all statistical analysis was performed in Prism and statistical significance was indicated. In this revised version, we have hopefully provided sufficient details for the reviewers and readers to appreciate our rigor in analyzing the data from our many experiments.

Before submission of the first version, we had collected all the data prior to cropping for western blots and for images that were presented throughout the paper. All the blots were originally submitted as a large PDF. The revised submission now has improved labelling of these back-up files.

4. Is the manuscript presented in an intelligible fashion and written in standard English?

Reviewer #1: Yes

Reviewer #2: Yes

5. Review Comments to the Author

Reviewer #1: In their manuscript, Yee et al report that after status epilepticus (SE), an orchestrated metabolic change occurs which is governed by WNT-signaling and downstream mTOR activation, resulting in glycolysis shunting the Krebs cycle.

This paper is a very comprehensive report - commendably in many experiments based on two different SE models. Some issues, however, need to be clarified:

Authors’ Response. Thank you for your positive comments on the comprehensive nature of the work. This was a complex mechanism to craft to incorporate all our observations.

Major concerns:

1. Overall, there is a strong mix of methods and results, with methods often being elaborately explained in the results section or the figure legends (which were integrated in the text and often split, which made reading difficult). The authors should try to disentangle this mixture as far as possible to increase legibility.

2. Similarly, also results and introduction are mixed to some degree: Lines 110-112 and again 123-126 are somewhat summing up the results within the Introduction section. Again, this should be avoided.

3. The discussion is too long and should be condensed.

Authors’ response to major concerns 1, 2 and 3. We have hopefully revised the methods, results and discussion to improve readability. Because of new experiments (detailed in Figure 7) and our own realization of the elevated significance of the work, the results and discussion sections have been extensively revised, especially the latter. We attempted to shorten the discussion, but then needed to better highlight the metabolic implications. Because we added a genetic animal model of Wnt signaling (see below), we discovered that the Wnt signaling triggered role of PDH regulation may play a larger than expected role in epileptogenesis and could be part of general mechanisms, whether it be genetic or acquired. 

4. Regarding Wnt and SE, the findings of Gorter et al (214, in Neurobiology of Disease No. 62) should be acknowledged.

Authors’ response. We have now noted this reference. Thank you.

Minor issues:

Line 45: the sentence is a bit cryptic. Please rephrase.

Line 448: Data is plural, hence please use plural also in the verb.

Authors’ response. We have now corrected these sentences. Yes, we did know that “data” is a plural noun, so this escaped our grammar and typo check.

Reviewer #2: The authors demonstrate a transient activation of the Wnt-beta catenin pathway during the latent period following SE similar as found in some cancers. This line of research requires investigation but the results as presented are associative and the authors do not show causality. My comments are as follow:

Authors’ response. Thank you for your critical and important comments. Please see below for my detailed response. 

a. I would suggest toning down statements such as saying this is the first to show the complex signaling and metabolic changes underlying epileptogenesis as the authors do not show causality. Also, since a recent study by Wolff et al 2020 argues that the latent period may not even exist as subclinical seizures may continue during this period. This study potentially helps increase our understanding but without data showing that inhibition of Wnt-beta-catenin pathways leads to less spontaneous recurrent seizures following SE- these results remain an association. Especially since the changes are see only for one day. Further experiments are required to address this significant weakness of this study.

Authors’ response. We have revised or decreased the statements regarding the uniqueness. I’ll note that the metabolic analysis that was beautifully done by Sonnewald and others did not include this early period of epileptogenesis, but instead focused on the ~14 day period after the initiating events and during the onset and establishment of chronic seizures with the pilocarpine model in the FVB mice.

The paper by Wolff et al was very useful, as it demonstrated hyperoxia in the hippocampus within the same timeframe that we observed aerobic glycolysis in our work. Secondly, epileptiform events were centered around day 5 of epileptogenesis, also consistent with our observations. In the Wolff study, kainate was administered directly into the brain. The pharmacokinetics and pharmacodynamics of kainate delivery are likely different than our studies, where kainate was delivered intraperitoneally. This difference complicates the meaning of any comparisons.

b. It is argued that Wnt signaling inhibitors do not cross the BBB and therefore their effect was not tested or when tested was ineffective. Why not deliver these drugs icv? Especially since 2DG was ineffective in normalizing the SE Induced changes. I think this is a critical missing part of this study.

c. To increase the impact of this study it would be necessary to treat the animals with wnt-beta-catenin inhibitors and demonstrate an effect on spontaneous recurrent seizures.

Authors’ response. We appreciate the reviewer’s thoughtful suggestions and agree that a direct link of Wnt signaling to the events observed in this paper is important. And we grappled with functionality or our mechanistic framework throughout this work. We attempted to use of Wnt pathway inhibitors (by intraperitoneal injection) as the simplest experimental path. We used XAV939 (tankyrase inhibitor; directed at the Axin-�-catenin destruction complex) or LG974 (porcupine inhibitor; directed at synthesis of all Wnt ligands). Neither drug gave convincing results for day 5 of epileptogenesis, likely by not crossing the blood brain barrier. XAV939 has been used in a 2019 PLOS 1 study with a reported consequence to dendrite outgrowth at day 14 or so of epileptogenesis. This study used intrahippocampal injection of kainate, but intraperitoneal injection of XAV939. This study used doses of XAV939 that had been effective in an animal model of pulmonary fibrosis, which would not require brain distribution. To my knowledge, the Porcupine inhibitor has been excluded from clinical trials with brain metastases because it does not penetrate to the brain.

While the reviewer’s suggestion of using an ICV injection is an excellent idea, it is particularly difficult at this time. In this period disrupted by the covid 19 pandemic, the proposed experimental direction is extremely difficult. We have neither the equipment nor expertise to do these ICV injections. This requires a new animal protocol amendment, which at our institution now takes months for mandatory review. Furthermore, the training to do ICV injections, if we could find the equipment, is difficult due to required and necessary social distancing. Both the training and animal infrastructure at our institution has been affected by the ongoing pandemic. Furthermore, suggested approach would require multiple experiments for determination of possible brain toxicity and statistical reproducibility since there is no precedence for XAV939 or LG974 to be delivered in this manner. Our study is amongst the first studies to ponder the role of Wnt signaling in epileptogenesis, so we do not know of prior studies to guide a dose-response, even if we could learn the technical aspects of a stereotactic injections. While this would be a good proof-of principle experiment for the future, ICV would not be the mode of delivery in human patients. We are currently investigating brain-penetrant therapeutic combinations aimed at blocking Wnt signaling, but these results will be ready for publication in the not too distant future.

But the reviewers’ question is a good one. In attempting to discern whether Wnt signaling has a causal role in epileptogenesis, we took a more difficult genetic approach in the new figure 7, generated by a re-examination of some completed, albeit unpublished findings. We thank the reviewer for prompting a re-interpretation of near-completed work. The collective interpretation is that constitutive Wnt signaling increases basal metabolic enzymes and mTOR. Furthermore, constitutive Wnt signaling should decrease the threshold for susceptibility to induced seizures, and in fact does so. In the end, the new (old) findings advance the role of Wnt signaling in epileptogenesis using the mechanisms advanced in this manuscript. 

Some background is necessary to interpret the new findings. To address the role of Wnt signaling in the biochemical and seizure changes, we used a mouse in which the HBP1 gene was deleted. We and others have shown that HBP1 is a transcriptional repressor that inhibits Wnt signaling. The mechanism is an inhibition of LEF/TCF transcriptional regulation (which is triggered by the upstream Wnt signaling pathway). We and others have shown that reduced levels of the HBP1 gene are associated with a poor prognosis in breast and other cancers. Because of the cancer correlation with Wnt signaling and HBP1, we sought to create a deletion of the HBP1 gene. Because of the known effect of mouse backgrounds on tumors, we carefully created strains in defined backgrounds C57BL6 and FVBN for future tumor work. These reagents are part of my lab work and were intended for our breast cancer studies. The disappointing result for the cancer studies was that the HBP1 knockouts had no spontaneous mammary tumors (though they exhibited colon cancers).

Nonetheless, a surprising observation for this paper was that these HBP1-/- mice had spontaneous seizures, discovered serendipitously by our animal facility. The work of the present manuscript also showed that HBP1 decreased in early epileptogenesis, coinciding with a peak in Wnt signaling. A deletion of HBP1 exhibited increased constitutive Wnt signaling in several tissues, as would be expected by the loss of an inhibitor. Only one previous report had linked HBP1 to brain cortical differentiation. Thus we re-examined some published observations and added new experiments to create a new Figure 7. Using a genetic argument and tools, the collective conclusions of Figure 7 are that HBP1 and Wnt signaling may be part of mechanisms that set the threshold for acquired seizure susceptibility: 1) HBP1 gene expression occurs in the CA1 and DG region known to associated with regulation of seizures; 2) The deletion of HBP1 in the hippocampus results in increased Wnt signaling, mTOR activation, and metabolic rearrangements. In particular, the induction of PDK4 and PDH phosphorylation are shared in our elaborated mechanisms and in the HBP1 deletion; 3) the HBP1+/- and HBP1-/- mice have heightened sensitivity to seizures induced by pilocarpine in kainate in the respective mouse backgrounds, 

The similarity in mechanisms between the pilocarpine and the HBP1-/- models prompted a realization that the discovered mechanisms from Wnt signaling-to-metabolism-to mTOR may be a fundamental discovery. In revising the discussion, we focused on the central role of PDH regulation and function and reviewed the literature on the very interesting role of PDH in genetic epilepsies. By obtaining the new data in Figure 7, we were now able to elevate the significance of metabolic transitions in epilepsies. We highlight in the revised manuscript that Wnt signaling is an important input into regulating PDH, which can contribute to epileptogenic, whether biochemically or genetically.

Thus, addressing the comments of the reviewers created an improved manuscript with more general and significant conclusions. We were surprised at the similarity of mechanisms, regardless of model used, and at the very central role of regulating PDH and metabolic transitions. Thus, we express our appreciation to the reviewers for their comments and thought to improve the paper.

d. I don’t clearly see the link between the mTOR and GABA story. Please remove the GABA and glutamate results which can be part of another paper.

Authors’ response. Yes, we do appreciate the concern of the reviewer. We do not report a relationship between mTOR and GABA. We have revised the manuscript to be absolutely sure there that a reader would not have this mistaken impression. The change in neurotransmitter homeostasis is an important aspect of this paper and feeds directly from the anaplerotic reactions of the metabolic rearrangement. Our data is most consistent with the known decrease in GABA. We show evidence that the GABA decline results from changes in metabolic homeostasis. Our data indicates that the complex glutamine-glutamate-GABA metabolic mechanisms and altered homeostasis is derived in part from the induced Warburg effect. An unappreciated and less discussed aspect of a Warburg metabolism are the anaplerotic mechanisms known to exist at the TCA intermediate alpha-ketoglutarate-to-glutamate. By systematically investigating the enzymes that contribute to glutamine/glutamate/GABA equilibrium in the brain, we observed a re-setting to new levels at day 5 that was fully consistent with the observed decrease in GABA levels. Such a mechanism fits plausibly in this period, but it is more complex than we had expected. We have attempted to further clarify the explanation. We feel that this complex mechanism now fits functionally into this paper.

e. Methods (line 944): Please explain further what is meant by “ a minimum of 4 photographic field…per mouse). For example, how many sections does this equate to. Was there attempts at a co-localization analysis? Was the person doing the counts blinded? Also please state how thick were your sections.

Authors’ response. We have clarified the quantitation on the figure legends and methods. From the revised methods

Confocal Imaging: Stained sections were imaged on a LSM800 laser scanning confocal microscope (ZEN 2.3-blue edition) in multi-tracking mode. The images were obtained with optimal pinhole a 63X, 1.4 NA, plan-apochromat oil immersion objective, 0.8 micron per optical sections, at a scan speed of 3, averaging of 4 images with an overall pixel dwell time of ~12 µsec. All image processing and analysis was performed in Fiji. In all images, only balance, contrast, and evenness of the illumination were adjusted. Final immunolabeling quantification and co-localization results were reported as percentage of positive cells (neuronal or astrocytes)/ total cells (neuronal or astrocytes). The quantification was conducted by three blinded investigators in four-six hippocampal coronal sections (section width= 12 µm) per mouse. Then, a minimum of 4 photographic fields were quantified per section (n = 3-4 mice for each group) using Fiji.

f. Methods Confocal (lines 942 and 945): Fiji and Image J are the same programs except the former has many already installed plugins.

Authors’ response. We apologize for the confusion. Fiji was used in all confocal images used in the figures.

g. Abstract: The following is repetitive and can be shortened “The results of this paper build a new molecular framework of complex signaling pathways for early epileptogenesis. Our framework advances the understanding of the complex molecular changes in early epileptogenesis and creates a detailed platform for discovering new future therapeutic strategies based upon Wnt signaling and/or metabolic interventions. Our studies may define a unique window in early epileptogenesis for attenuating recurrent and chronic seizures that define epilepsies.”

Authors’ response. We have re-written the abstract to make it more readable and to incorporate the new genetic angle. The use of two different experimental systems that yielded similar results additionally allowed some conclusions to be strengthened.

h. Introduction: Rapamycin also reduced seizures in a model of infantile spasms. Please add reference.

Authors’ response. Thank you. The reference has been added.

i. Introduction: May be useful to add figure which summarizes the relationship between Wnt signaling and mTOR.

Authors’ response. We have created a new Figure 1A to clarify the mechanisms.

j. Introduction: Frizzled receptors are mentioned without context.

Authors’ response. The text has been altered accordingly.

k. There are many abbreviations that require definition. Please provide a definition at their first appearance.

Authors’ response. We have revised the manuscript to insure that abbreviations are defined.

l. Introduction (line 114). Warburg metabolism is mentioned without a definition.

m. Introduction: (lines 115-116) …should say “that occurs in neurons” and not “…are occurs…”

Authors’ response. Thank you. We have revised the manuscript accordingly.

6. PLOS authors have the option to publish the peer review history of their article (what does this mean?). If published, this will include your full peer review and any attached files.

Do you want your identity to be public for this peer review? For information about this choice, including consent withdrawal, please see our Privacy Policy.

Reviewer #1: No

Reviewer #2: No

---

## [Decision Letter · Decision Letter 1]

14 May 2021

A Warburg-like Metabolic Program Coordinates Wnt, AMPK and mTOR Signaling Pathways in Epileptogenesis

PONE-D-20-22075R1

Dear Dr. Yee,

We’re pleased to inform you that your manuscript has been judged scientifically suitable for publication and will be formally accepted for publication once it meets all outstanding technical requirements. Please, check and fix "Student" in the final version, as the capital letter is required.

Kind regards,

Giuseppe Biagini, MD

Academic Editor

PLOS ONE

Additional Editor Comments (optional):

Reviewers' comments:

Reviewer's Responses to Questions

**Comments to the Author**

1. If the authors have adequately addressed your comments raised in a previous round of review and you feel that this manuscript is now acceptable for publication, you may indicate that here to bypass the “Comments to the Author” section, enter your conflict of interest statement in the “Confidential to Editor” section, and submit your "Accept" recommendation.

Reviewer #1: All comments have been addressed

Reviewer #2: All comments have been addressed

2. Is the manuscript technically sound, and do the data support the conclusions?

Reviewer #1: Yes

Reviewer #2: Yes

3. Has the statistical analysis been performed appropriately and rigorously? 

Reviewer #1: Yes

Reviewer #2: Yes

4. Have the authors made all data underlying the findings in their manuscript fully available?

Reviewer #1: Yes

Reviewer #2: Yes

5. Is the manuscript presented in an intelligible fashion and written in standard English?

Reviewer #1: Yes

Reviewer #2: Yes

6. Review Comments to the Author

Reviewer #1: Many thanks for the extensive revision and indeed the new experiments, which certainly make the case even stronger.

Reviewer #2: (No Response)

7. PLOS authors have the option to publish the peer review history of their article (what does this mean?). If published, this will include your full peer review and any attached files.

Reviewer #1: No

Reviewer #2: No

---

## [Editor Report · Acceptance letter]

22 Jul 2021

PONE-D-20-22075R1 

A Warburg-like Metabolic Program Coordinates Wnt, AMPK, and mTOR Signaling Pathways in Epileptogenesis 

Dear Dr. Yee:

I'm pleased to inform you that your manuscript has been deemed suitable for publication in PLOS ONE. Congratulations! Your manuscript is now with our production department. 

Kind regards, 

on behalf of

Dr. Giuseppe Biagini 

Academic Editor

PLOS ONE